# Differentiable Multiple Shooting Layers

**Stefano Massaroli**\*
The University of Tokyo, `DiffEqML`
`massaroli@robot.t.u-tokyo.ac.jp`

**Michael Poli**\*
KAIST, `DiffEqML`
`poli_m@kaist.ac.kr`

**Sho Sonoda**
RIKEN

**Taiji Suzuki**
The University of Tokyo, RIKEN

**Jinkyoo Park**
KAIST

**Atsushi Yamashita**
The University of Tokyo

**Hajime Asama**
The University of Tokyo

## Abstract

We detail a novel class of implicit neural models. Leveraging time–parallel methods for differential equations, *Multiple Shooting Layers* (MSLs) seek solutions of initial value problems via parallelizable root-finding algorithms. MSLs broadly serve as drop–in replacements for *neural ordinary differential equations* (Neural ODEs) with improved efficiency in number of function evaluations (NFEs) and wall–clock inference time. We develop the algorithmic framework of MSLs, analyzing the different choices of solution methods from a theoretical and computational perspective. MSLs are showcased in long horizon optimal control of ODEs and PDEs and as latent models for sequence generation. Finally, we investigate the speedups obtained through application of MSL inference in neural controlled differential equations (Neural CDEs) for time series classification of medical data.

## 1   Introduction

*For the last twenty years, one has tried to speed up numerical computation mainly by providing ever faster computers. Today, as it appears that one is getting closer to the maximal speed of electronic components, emphasis is put on allowing operations to be performed in parallel. In the near future, much of numerical analysis will have to be recast in a more "parallel" form. Nievergelt, 1964*

Discovering and exploiting parallelization opportunities has allowed deep learning methods to succeed across application areas, reducing iteration times for architecture search and allowing scaling to larger data sizes (Krizhevsky et al., 2012; Diamos et al., 2016; Vaswani et al., 2017). Inspired by multiple shooting, time–parallel methods for ODEs (Bock and Plitt, 1984; Diehl et al., 2006; Gander, 2015; Staff and Rønquist, 2005) and recent advances on the intersection of differential equations, implicit problems and deep learning, we present a novel class of neural models designed to maximize parallelization across *time*: differentiable *Multiple Shooting Layers* (MSLs). MSLs seek solutions of *initial value problems* (IVPs) as roots of a function designed to ensure satisfaction of boundary constraints. Figure 1 provides visual intuition of the parallel nature of MSL inference.

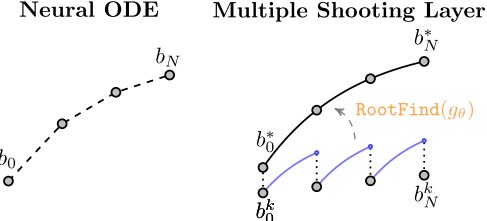

Figure 1: MSLs apply parallelizable root finding methods to obtain differential equation solutions.

---

\*Equal contribution. Author order was decided by flipping a coin.

35th Conference on Neural Information Processing Systems (NeurIPS 2021).

■ **An implicit neural differential equation**   MSL inference is built on the interplay of numerical methods for root finding problems and differential equations. This property reveals the proposed method as a missing link between implicit–depth architectures such as Deep Equilibrium Newtorks (DEQs) (Bai et al., 2019) and continuous–depth models (Weinan, 2017; Chen et al., 2018; Massaroli et al., 2020; Kidger et al., 2020b; Li et al., 2020). Indeed, MSLs can be broadly applied as drop–in replacements for Neural ODEs, with the advantage of often requiring a smaller *number of function evaluations* (NFEs) for neural networks parametrizing the vector field. MSL variants and their computational signature are taxonomized on the basis of the particular solution algorithm employed, such as Newton and *parareal* (Maday and Turinici, 2002) methods.

■ **Faster inference and fixed point tracking**   Differently from classical multiple shooting methods, MSLs operate in regimes where function evaluations of the vector field can be significantly more expensive than surrounding operations. For this reason, the reduction in NFEs obtained through time–parallelization leads to significant inference speedups. In full-batch training regimes, MSLs provably enable tracking of fixed points across training iterations, leading to drastic acceleration of forward passes (often the cost of a single root finding step). We apply the tracking technique to optimal control of ODEs and PDEs, with speedups in the order of several times over Neural ODEs. MSLs are further evaluated in sequence generation via a latent variant, and as a faster alternative to *neural controlled differential equations* (Neural CDE) (Kidger et al., 2020b) in long–horizon time series classification.

## 2   Multiple Shooting Layers

Consider the *initial value problem* (IVP)

$$\begin{aligned} \dot{z}(t) &= f_\theta(t, z(t)) \\ z(0) &= z_0 \end{aligned}, \quad t \in [0, T]. \tag{2.1}$$

with state $z \in \mathcal{Z} \subset \mathbb{R}^{n_z}$, parameters $\theta \in \mathcal{W}$ for some space $\mathcal{W}$ of functions $[0, T] \to \mathbb{R}^{n_\theta}$ and a smooth vector field $f_\theta : [0, T] \times \mathcal{Z} \times \mathcal{W} \to \mathcal{Z}$. For all $z \in \mathcal{Z}$, $s, t \in [0, T]; s < t$ we denote with $\phi_\theta(z, s, t)$ the solution of (2.1) at time $t$ starting from $z$ at time $s$, i.e. $\phi_\theta(z, s, t) : (z, s, t) \mapsto z(t)$.

The crux behind *multiple–shooting* methods for differential equations is to turn the initial value problem (2.1) into a *boundary value problem* (BVP). We split the the time interval $[0, T]$ in $N$ sub–intervals $[t_n, t_{n+1}]$ with $0 = t_0 < t_1 < \cdots < t_N = T$ and define $N$ left boundary subproblems

$$z_n(t_n) = b_n \text{ and } \dot{z}_n(t) = f_\theta(t, z_n(t)), \ t \in [t_n, t_{n+1}] \tag{2.2}$$

where $b_n$ are denoted as *shooting parameters*. At each time $t \in [0, T]$, the solution of (2.2) matches the one of (2.1) iff all the shooting parameters $b_n$ are identical to $z(t_n)$, $b_n = \phi_\theta(z_0, t_0, t_n)$. Using $z(t_n) = \phi_\theta(z(t_{n-1}), t_{n-1}, t_n)$, we obtain the equivalent conditions

$$\begin{aligned} b_0 &= \phi_\theta(z_0, t_0, t_0) = z_0 \\ b_1 &= \phi_\theta(b_0, t_0, t_1) = z_0(t_1) \\ &\vdots \\ b_N &= \phi_\theta(b_{N-1}, t_{N-1}, t_N) = z_{N-1}(t_N) \end{aligned}$$

Let $B := (b_0, b_1, \cdots, b_N)$ and $\gamma_\theta(B, z_0) := (z_0, \phi_\theta(b_0, t_0, t_1), \cdots, \phi_\theta(b_{N-1}, t_{N-1}, t_N))$. We can thus turn the IVP (2.1) into the roots–finding problem the of a function $g_\theta$ defined as

$$g_\theta(B, z_0) = B - \gamma_\theta(B, z_0)$$

**Definition 1** (Multiple Shooting Layer (MSL)). *With $\ell_x : \mathcal{X} \to \mathcal{Z}$ and $\ell_y : \mathcal{Z}^{N+1} \to \mathcal{Y}$ two affine maps, a* `multiple shooting layer` *is defined as the implicit input–output mapping* $x \mapsto y$:

$$\begin{aligned} z_0 &= \ell_x(x) \\ B^* &: g_\theta(B^*, z_0) = \mathbb{0} \\ y &= \ell_y(B^*) \end{aligned} \tag{2.3}$$

# 3 Realization of Multiple Shooting Layers

The remarkable property of MSL is the possibility of computing the solutions of all the $N$ IVPs (2.2) in parallel from the shooting parameters in $B$ with any standard ODE solver. This allows for a drastic reduction in the number of vector field evaluations at the cost of a higher memory requirement for the parallelization to take place. Nonetheless, the forward pass of MSLs requires the shooting parameters $B$ to satisfy the nonlinear algebraic matching condition $g_\theta(B, z_0) = \mathbb{0}$, which has also to be solved numerically.

## 3.1 Forward Model

The *forward* MSL model involves the synergistic combination of two main classes of numerical methods: *ODE solvers* and *root finding* algorithms, to compute $\gamma_\theta(B, z_0)$ and $B^*$, respectively. There exists a hierarchy between the two classes of methods: the ODE solver will be invoked at each step $k$ of the root finding algorithm to compute $\gamma_\theta(B^k, z_0)$ and evaluate the matching condition $g_\theta(B^k, z_0)$.

**Newton methods for root finding**   Let us denote with $B^k$ the solution of the root finding problem at the $k$-th step of the Newton method and let $\mathsf{D}g_\theta(B^k, z_0)$ be the Jacobian of $g_\theta$ computed in $B^k$. The solution $B^*\ :\ g_\theta(B^*, z_0) = \mathbb{0}$ can be obtained by iterating the Newton–Raphson fixed point iteration

$$B^{k+1} = B^k - \alpha \left[ \mathbb{1}_N \otimes \mathbb{1}_{n_z} - \mathsf{D}\gamma_\theta(B^k, z_0) \right]^{-1} \left[ B^k - \gamma_\theta(B^k, z_0) \right] \tag{3.1}$$

which converges quadratically to $B^*$ (Nocedal and Wright, 2006). The exact Newton iteration theoretically (3.1) requires the inverse of the Jacobian $\mathbb{1}_N \otimes \mathbb{1}_{n_z} - \mathsf{D}\gamma_\theta(B^k, z_0)$. Without the special structure of the MSL problem, the Jacobian would have had to be the computed in full, as in the case of DEQs (Bai et al., 2019). Being the Jacobian of dimension $\mathbb{R}^{Nn_z \times Nn_z}$, its computation with *reverse–mode* automatic differentiation (AD) tools scales poorly with state dimensions and number of shooting parameter (cubically in both $n_z$ and $N$).

Instead, the special structure of the MSL matching function $g_\theta(B, z_0) = B - \gamma_\theta(B, z_0)$ and its Jacobian, opens up application of direct updates where inversion is not required.

**Direct multiple shooting**   Following the treatment of Chartier and Philippe (1993), we can obtain a *direct* formulation of the Newton iteration which does not require the composition of the whole Jacobian nor its inversion. The direct multiple shooting iteration is derived by setting $\alpha = 1$ and multiplying the Jacobian on both sides of (3.1) yielding

$$\left[ \mathbb{1}_N \otimes \mathbb{1}_{n_z} - \mathsf{D}\gamma_\theta(B^k, z_0) \right] (B^{k+1} - B^k) = \gamma_\theta(B^k, z_0) - B^k$$

which leads to the following update rule for the individual shooting parameters $b_n^k$ (see Fig. 2):

$$b_{n+1}^{k+1} = \phi_{\theta,n}(b_n^k) + \mathsf{D}\phi_{\theta,n}(b_n^k) \left( b_n^{k+1} - b_n^k \right), \ \ b_0^{k+1} = z_0 \tag{3.2}$$

where $\mathsf{D}\phi_{\theta,n}(b_n^k) = \mathrm{d}\phi_{\theta,n}(b_n^k)/\mathrm{d}b_n$ is the sensitivity of each individual flow to its initial condition. Due to the dependence of $b_{n+1}^{k+1}$ on $b_n^{k+1}$, a complete Newton iteration theoretically requires $N - 1$ sequential stages.

**Finite-step convergence**   Iteration (3.2) exhibits convergence to the exact solution of the IVP (2.1) in $N - 1$ steps (Gander, 2018, Theorem 2.3). In particular, given perfect integration of the sub–IVPs, $b_n^k$ coincides with the exact solution $\phi_\theta(z_0, t_0, t_n)$ from iteration index $k = n$ onward, i.e. at iteration $k$ only the last $N - k$ shooting parameters are actually updated. Thus, the computational and memory footprint

**Single Stage of Newton Iteration**

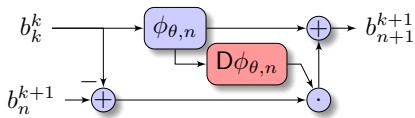

Figure 2: One stage of Newton iteration (3.2).

of the method diminishes with the number of iterations. This result can be visualized in the graphical representation of iteration (3.2) in Figure 3 while further details are discussed in Appendix B.1.

**Time–Iteration Propagation of Newton Scheme**

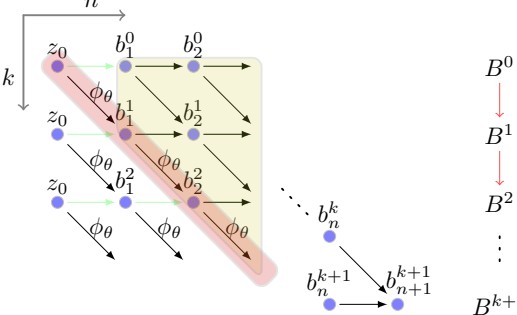

Figure 3: Propagation in $k$ and $n$ of the Newton iteration (3.2). The intertwining between the updates in $n$ and $k$ leads to the finite step convergence result. In fact, by setting $b_0^0 = z_0$, we see how the correcting term $b_n^{k+1} - b_n^k$ multiplying the flow sensitivity $\mathsf{D}\phi_{\theta,n}$ progressively nullifies at the same rate in $n$ and $k$. As a result, the exact sequential solution of the IVP (2.1) unfolds on the diagonal $k = n$ and the only *active* part of the algorithm is the one above the diagonal (highlighted in yellow).

**Numerical implementation** Practical implementation of the Newton iteration (3.2) requires an ODE solver to approximate the flows $\phi_{\theta,n}(b_n^k)$ and an algorithm to compute their sensitivities w.r.t. $b_n^k$. Besides direct application of AD, we show an efficient alternative to obtain all $\mathsf{D}\phi_{\theta,n}$ in parallel alongside the flows, with a single call of the ODE solver.

**Efficient exact sensitivities** Differentiating through the steps of the forward numerical ODE solver using reverse–mode AD is straightforward, but incurs in high memory cost, additional computation to *unroll* the solver steps and introduces further numerical error on $\mathsf{D}\phi_{\theta,n}$. Even though the memory footprint might be mitigated by applying the *adjoint* method (Pontryagin et al., 1962), this still requires to solve backward the $N - k$ adjoint ODEs and sub–IVPs (2.2), at each iteration $k$. We leverage *forward* sensitivity analysis to compute $\mathsf{D}\phi_{\theta,n}$ alongside $\phi_{\theta,n}$ in a single call of the ODE solver. This approach, which might be considered as the continuous variant of forward–mode AD, scales quadratically with $n_z$, has low memory cost, and explicitly controls numerical error.

**Proposition 1** (Forward Sensitivity (Khalil, 2002)). *Let $\phi_\theta(z, s, t)$ be the solution of* (2.1). *Then,* $v(t) = \mathsf{D}\phi_\theta(z, s, t)$ *satisfies the linear matrix–valued differential equations*

$$\dot{v}(t) = \mathsf{D}f_\theta(t, z(t))v(t), \quad v(s) = \mathbb{I}_{n_z} \quad \text{where } \mathsf{D}f_\theta \text{ denotes } \partial f_\theta / \partial z.$$

Therefore, at iteration $k$ all $\mathsf{D}\phi_{\theta,n}(b_n^k)$ can be computed in parallel while performing the forward integration of the $N - k$ IVPs (2.2) and their forward sensitivities, i.e.

$$\texttt{ForwardSensitivity}: \ \left\{ b_n^k \mapsto (\phi_{\theta,n}, \mathsf{D}\phi_{\theta,n}) \right\}_{k < n \leq N}$$

which enables full *vectorization* of Jacobian–matrix products between $\partial f_\theta / \partial z$ and $v$ as well as maximizing re–utilization of vector field evaluations. Detailed derivations are provided in Appendix B.2. Appendix C.1 analyzes practical considerations and software implementation of the algorithm.

**Zero–order approximate iteration** In high state dimension regimes, the quadratic memory scaling of the forward sensitivity method might be infeasible. If this is the case, a zero–order approximation of the Newton iteration preserving the finite–step converge property can be employed: the *parareal* method Lions et al. (2001). From the Taylor expansion of $\phi_{\theta,n}(b_n^{k+1})$ around $b_n^k$

$$\phi_{\theta,n}(b_n^{k+1}) = \phi_{\theta,n}(b_n^k) + \mathsf{D}\phi_{\theta,n}(b_n^k) \left( b_n^{k+1} - b_n^k \right) + o \left( \| b_n^{k+1} - b_n^k \|_2^2 \right),$$

we have the following approximant for the correction term of (3.2)

$$\mathsf{D}\phi_{\theta,n}(b_n^k) \left( b_n^{k+1} - b_n^k \right) \approx \phi_{\theta,n}(b_n^{k+1}) - \phi_{\theta,n}(b_n^k). \tag{3.3}$$

*Parareal* computes the RHS of (3.3) by *coarse*[2] numerical solutions $\psi_{\theta,n}(b_n^k)$, $\psi_{\theta,n}(b_n^{k+1})$ of $\phi_{\theta,n}(b_n^k)$, $\phi_{\theta,n}(b_n^k)$, leading to the forward iteration,

$$b_{n+1}^{b+1} = \phi_{\theta,n}(b_n^k) + \psi_{\theta,n}(b_n^{k+1}) - \psi_{\theta,n}(b_n^k).$$

---

[2]e.g. few steps of a low–order ODE solver

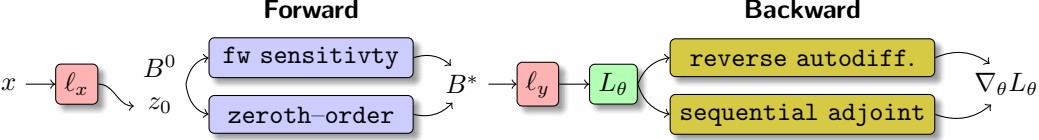

**Forward**        **Backward**

Figure 4: Scheme of the forward–backward pass of MSLs. After applying the input map $\ell_x$ to the input $x$ and choosing initial shooting parameters $B^0$, the forward pass is iteratively computed with one of the numerical schemes described in Sec. 3.1 which, in turn, makes use of some ODE solver to compute $\phi_{\theta,n}$ in parallel, at each step. Once the output $y$ and the loss are computed computed by applying $\ell_y$ and $L_\theta$ to $B^*$, the loss gradients can be computed by standard adjoint methods or reverse–mode automatic differentiation.

## 3.2 Properties of MSLs

**Differentiating through MSL**  Computing loss gradients through MSLs can be performed by directly back–propagating through the steps of the numerical solver via reverse–mode AD.

A memory efficient alternative is to apply the sequential adjoint method to the underlying Neural ODE. In particular, consider a loss functions computed independently with the values of different shooting parameters, $L(x, B^*, \theta) = \sum_{n=1}^{N} c_\theta(x, b_n^*)$. The adjoint gradient for the MSL is then given by

$$\nabla_\theta L = \int_0^T \lambda^\top(t) \nabla_\theta f_\theta(t, z(t)) \mathrm{d}t$$

where the Lagrange multiplier $\lambda(t)$ satisfies a backward piecewise–continuous linear ODE

$$\dot{\lambda}(t) = -\mathsf{D}f_\theta(t, z(t))\lambda(t) \qquad \text{if } t \in [t_n, t_{n+1})$$
$$\lambda^-(t_n) = \lambda(t_n) + \nabla_b^\top c_\theta(x, b_n) \qquad \lambda(T) = \nabla_b^\top c_\theta(x, b_N)$$

The adjoint method typically requires the IVP (2.1) to be solved backward alongside $\lambda$ to retrieve the value of $z(t)$ needed to compute the Jacobians $\mathsf{D}f_\theta$ and $\nabla_\theta f_\theta$. This step introduces additional errors on the final gradients: numerical errors accumulated on $b_N^* \approx z(T)$ during forward pass, propagate to the gradients and sum up with errors on the backward integration of (2.1).

Here we take a different, more robust direction by interpolating the shooting parameters and drop the integration of (2.1) during the backward pass. The values of the shooting parameters retrieved by the forward pass of MSLs are solution points of the IVP (2.1), i.e. $b_n^* = \phi(z_0, t_0, t_n)$ (up to the forward numerical solver tolerances). On this assumption, we construct a `cubic spline` interpolation $\hat{z}(t)$ of the shooting parameters $b_n^*$ and we query it during the integration of $\lambda$ to compute the Jacobians of $f_\theta$. Further results on back–propagation of MSLs are provided in Appendix B.3. Appendix C.4 practical aspects of the backward model alongside software implementation of the *interpolated* adjoint. The entire scheme of a forward–backward pass of an MSL is shown in Figure 4.

**One-step inference: fixed point tracking**  Consider training a MSL to minimize a twice–differentiable loss function $L(x, B^*, \theta)$ with Lipschitz constant $m_L^\theta$ through the gradient descent iteration

$$\theta_{p+1} = \theta_p - \eta_p \nabla_\theta L(x, B_p^*, \theta_p)$$

where $\eta_p$ is a positive learning rate and $B_p^*$ is the exact root of the matching function $g_\theta$ computed with parameters $\theta_p$ (i.e. the exact solution of the IVP (2.1) at the boundary points). Due to Lipschitzness of $L$, we have the following uniform bound on the variation of the parameters across training iterations

$$\|\theta_{p+1} - \theta_p\|_2 \le \eta_p m_L^\theta.$$

If we also assume $\gamma_\theta$ to be Lipschitz continuous w.r.t $z$ and $\theta$ with constants $m_\gamma^\theta$, $m_\gamma^z$ and differentiable w.r.t. $\theta$ we can obtain the variation of the fixed point $B^*$ to small changes in the model parameters by linearizing solutions around $\theta_p$

$$B_{p+1}^* - B_p^* = [\theta_{p+1} - \theta_p]\frac{\partial \gamma_{\theta_p}(B_p^*, z_0)}{\partial \theta} + o(\|\theta_{p+1} - \theta_p\|_2^2)$$

to obtain the uniform bound

$$\|B_{p+1}^* - B_p^*\|_2 \le \eta_p m_L^\theta m_\gamma^\theta + o(\|\theta_{p+1} - \theta_p\|_2^2).$$

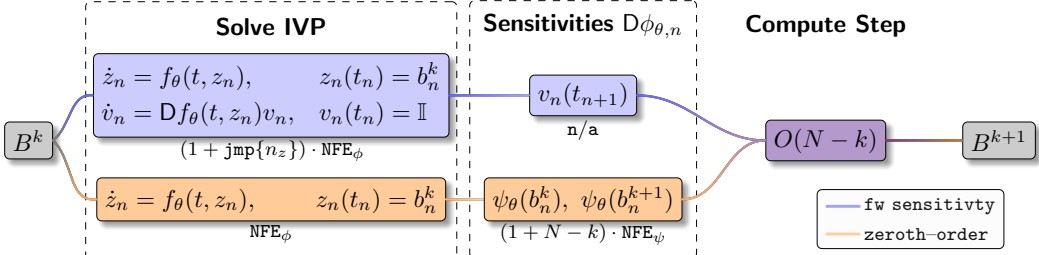

Figure 5: Single iteration computational *span* (McCool et al., 2012) in MSL. We normalize to 1 the cost of evaluating $f_\theta$. The $N - k$ sub–IVPs are solved in parallel in their sub–intervals, thus requiring a minimum span $\texttt{NFE}_\phi$. *Forward sensitivity* introduces Jacobian–matrix products costs amounting to $\texttt{jmp}$, which can be further parallelized into $\texttt{jvps}$.

Having a bounded variation on the solutions of the MSL for small changes of the model parameters $\theta$, we might think of *recycling* the previous shooting parameters $B_p^*$ as an initial guess for the direct Newton algorithms in the forward pass succeeding the gradient descent update. We show that by choosing a sufficiently small learning rate $\eta_p$ one Newton iteration can be sufficient to track the true value of $B^*$ during training. In particular, the following bound can be obtained.

**Theorem 1** (Quadratic fixed-point tracking). *If $f_\theta$ is twice continuously differentiable in $z$ then*

$$\|B_{p+1}^* - \bar{B}_p^*\|_2 \leq M\eta_p^2 \tag{3.4}$$

*for some $M > 0$. $\bar{B}_p^*$ is the result of one Newton iteration applied to $B_p^*$.*

The proof, reported in Appendix A.1, relies on the quadratic converge of Newton method. The quadratic dependence of the tracking error bound on $\eta_k$ allows use of typical learning rates for standard gradient based optimizers to keep the error under control. In this way, we can turn the *implicit* learning problem into an explicit one where the implicit inference pass reduces to one Newton iteration. This approach leads to the following training dynamics:

$$\theta \leftarrow \theta - \eta\nabla_\theta L(x, B^*, \theta)$$
$$B^* \leftarrow \texttt{apply}\{(3.2), B^*\}$$

We note that the main limitation of this method is the assumption on input $x$ to be constant across training iterations (i.e. the initial condition $z_0$ is constant as well). If the input changes during the training (e.g. under mini-batch SGD training regime), the solutions of the IVP (2.1) and thus its corresponding shooting parameters may drastically change with $x$ even for small learning rates.

**Numerical scaling** Each class of MSL outlined in Section 3.1 is equipped with unique computational scaling properties. In the following, we denote with $\texttt{NFE}_\phi$ the total number of vector field $f_\theta$ evaluations done, in parallel across shooting parameters, in a single sub–interval $[t_n, t_{n+1}]$. Similarly, $\texttt{NFE}_\psi$ indicates the function evaluations required by the *coarse* solver used for *parareal* approximations. Here, we set out to investigate the computational signature of MSLs as parallel algorithms. To this end, we decompose a single MSL iteration into two core steps: solving the IVPs across sub–intervals and computing sensitivities $\text{D}\phi$ (or their approximation). Figure 5 provides a summary of the *algorithmic span*[3] of a single MSL iteration as a function of number of Jacobian vector products ($\texttt{jvp}$), Jacobian matrix products ($\texttt{jmp}$), and vector field evaluations ($\texttt{NFE}$).

Fw sensitivity MSL frontloads the cost of computing $\text{D}\phi_{\theta,n}$ by solving the forward sensitivity ODEs of Proposition 1 alongside the evaluation of $\gamma_\theta$. Forward sensitivity equations involve a $\texttt{jmp}$, which can be optionally further parallelized as $n_z$ $\texttt{jvps}$ by paying a memory overhead. Once sensitivities have been obtained along with $\gamma_\theta$, no additional computation needs to take place until application of the shooting parameter update formula. The forward sensitivity approach thus enjoys the highest degree of time–parallelizability at a larger memory cost.

Zeroth–order MSL computes $\gamma_\theta$ via a total of $\texttt{NFE}_\phi$ evaluations $f_\theta$, parallelized across sub–intervals. The cheaper IVP solution in both memory and compute is however counterbalanced during calculation

---

[3]Longest sequential cost, in terms of computational primitives, that is not parallelizable due to problem–specific dependencies. A specific example for MSLs are the sequential $\texttt{NFE}_\phi$ calls required by the sequential ODE solver for each sub–interval.

of the sensitivities, as this MSL approach approximates the sensitivities $D\phi_{\theta,n}$ by a zeroth–order update requiring $N - k$ sequential calls to a coarse solver.

The analysis of MSL backpropagation scaling is straightforward, as sequential adjoints for MSLs mirror standard sensitivity techniques for Neural ODEs in both compute and memory footprints. Alternatively, `AD` can be utilized to backpropagate through the operations of the forward pass methods in use. This approach introduces a non–constant memory footprint which scales in the number of forward iterations and thus depth of computational graph.

## 4 Applications

### 4.1 Variational MSL

Let $x : \mathbb{R} \to \mathbb{R}^{n_x}$, be an observable of some continuous–time process and let $X = \{x_{-M}, \ldots, x_0, \ldots, x_N\} \in \mathbb{R}^{(M+N+1)\times n_x}$ be a sequence of observations of $x(t)$ at time instants $t_{-M} < \cdots < t_0 < \cdots < t_N$. We seek a model able to predict $x_1, \ldots, x_N$ given past observations $x_{-M}, \ldots, x_0$, equivalent to approximating the conditional distribution $p(x_{1:N}|x_{-M:0})$. To this end we introduce *variational* MSLs ($v$MSLs) as the following latent variable model:

$$
\begin{aligned}
(\mu, \Sigma) &= \mathcal{E}_\omega(x_{-M:0}) & &\text{Encoder } \mathcal{E}_\omega \\
q_\omega(z_0|x_{-M:0}) &= \mathcal{N}(\mu, \Sigma) & &\text{Approx. Posterior} \\
z_0 &\sim q_\omega(z_0|x_{-M:0}) & &\text{Reparametrization}
\end{aligned}
\qquad
\begin{aligned}
B^* &: g_\theta(B^*, z_0) = \mathbb{0} & &\text{Decoder } \mathcal{D}_\theta \\
\hat{x}_1, \ldots, \hat{x}_N &= \ell(B^*) & &\text{Readout } \ell
\end{aligned}
$$

Once trained, such model can be also used to generate new realistic sequences of the observable $x(t)$ by querying the decoder network at a desired $z_0$. $v$MSLs are designed to scale data generation to longer sequences, exploiting wherever possible parallel computation in time in both encoder as well as decoder modules. The structure of $\mathcal{E}_\omega$ is designed to leverage modern advances in representation learning for time–series via temporal convolutions (TCNs) or attention operators (Vaswani et al., 2017) to offer a higher degree of parallelizability compared to sequential encoders e.g RNNs, ODE-RNNs (Rubanova et al., 2019) or Neural CDEs (Kidger et al., 2020b). This, in turn, allows the encoder to match the decoder in efficiency, avoiding unnecessary bottlenecks. The decoder $\mathcal{D}_\theta$ is composed of a MSL which is tasked to unroll the generated trajectory in latent space. $v$MSLs are trained via traditional likelihood methods. The iterative optimization problem can be cast as the maximization of an evidence lower bound (`ELBO`):

$$
\min_{(\theta,\omega)} \mathbb{E}_{z_0 \sim q_\omega(z_0|x_{-M:0})} \sum_{n=1}^{N} \log p_n(\hat{x}_n) - \text{KL}(q_\omega || \mathcal{N}(\mathbb{0}, \mathbb{1}))
$$

with $p_n(\hat{x}_n) = \mathcal{N}(x_n, \sigma_n)$ and the standard deviations $\sigma_n$ are left as a hyperparameters.

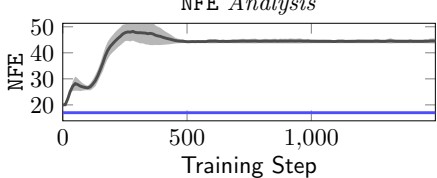

Figure 6: Mean and standard deviation of NFEs during vMSL and Latent Neural ODE across training trials. vMSLs require $60\%$ less NFEs during both training and inference.

**Sequence generation under noisy observations** We apply $v$MSLs on a sequence generation task given trajectories corrupted by state–correlated noise. We consider a baseline latent model sharing the same overall architecture as $v$MSLs, with a Neural ODE as decoder. In particular, the Neural ODE is solved via the `dopri5` solver with absolute and relative tolerances set to $10^{-4}$, whereas the $v$MSL decoder is an instance of `fw sensitivity MSL`. The encoder for both models is comprised of two layers of *temporal convolutions* (TCNs). All decoders unroll the trajectories directly in output space without additional readout networks. The validation on sample quality is then performed by inspection of the learned vector fields and the error against the nominal across the entire state–space. The proposed model obtains equivalent results as the baseline at a significantly cheaper computational cost. As shown in Figure 6, $v$MSLs require $60\%$ less NFEs for a single training iteration as well as for sample generation, achieving results comparable to standard Latent Neural ODEs (Rubanova et al., 2019). We report further details and results in Appendix E.1.

## 4.2 Neural Optimal Control

Beyond sequence generation, the proposed framework can be applied to *optimal control*. Here we can fully exploit the drastic computational advantages of MSLs. In fact, we leverage on the natural assumption of finiteness of initial conditions $z_0$ where the controlled system (or *plant*) is initialized to verify the result of Th. 1. Let us consider a controlled dynamical system

$$\dot{z}(t) = f(t, z(t), \pi_\theta(t, z)), \quad z(0) = z_0 \quad (4.1)$$

with a parametrized policy $\pi_\theta : t, z \mapsto \pi_\theta(t, z)$ and initial conditions $z_0$ ranging in a finite set $Z_0 = \{z_0^j\}_j$. We consider the problems of stabilizing a low–dimensional *limit cycle* and deriving an optimal boundary policy for a linear PDE discretized as a 200–dimensional ODE.

**Limit cycle stabilization** We consider a *stabilization* task where, given a desired closed curve $S_d = \{z \in \mathcal{Z} : s_d(z) = 0\}$, we minimize the 1-norm between the given curve and the MSL solution of (4.1) across the timestamps

### Neural Optimal Policy via MSLs

*Vector Field*  *Learned Controller*

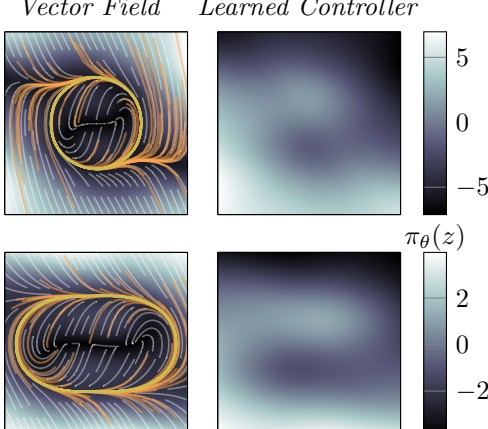

Figure 7: [Left] Closed–loop vector fields and trajectories corresponding to the $u_\theta$-controlled MSL. [Right] Learned controller $u_\theta(z)$ ($z \in \mathbb{R}^2$) for the two different desired limit cycles. Although, the inference of the all trajectory is performed with just two steps of RK4 (8 NFE), the initial accuracy of `dopri5` ($> 3000$ NFE) is preserved throughout training.

as well as the *control effort* $|\pi_\theta|$. We verify the approach on a one degree–of–freedom mechanical system, aiming with closed curves $S_d$ of various shapes. Following the assumptions on fixed point tracking and slow–varying–flows of Th. 1, we initialize $B_j^0$ by `dopri5` adaptive–step solver set with tolerances $10^{-8}$. Then, at each training iteration, we perform inference with a single parallel `rk4` step for each sub–interval $[t_n, t_{n+1}]$ followed by a single Newton update. Figure 7 shows the learned vector fields and controller, confirming a successful system stabilization of the system to different types of closed curves. We compare with a range of baseline Neural ODEs, solved via `rk4` and `dopri5`. Training of the controller via MSLs is achieved with orders of magnitude *less* wall–clock time and NFEs. Figure 8 shows the difference in NFEs w.r.t. `dopri5` while Fig. 9 compares wall–clock times per training iteration of MSL, `dopri5` and $rk4$.

We further provide *Symmetric Mean Average Percentage Error* (SMAPE) measurements between trajectories obtained via MSLs and an adaptive–step solver. MSLs initialized with recycled solutions are able to track the nominal trajectories across the entire training process. Additional details on the experimental setup, including wall–clock time comparisons with `rk4` and `dopri5` baseline Neural ODEs is are provided in Appendix E.2.

**Neural Boundary Control of the Timoshenko Beam** We further show how MSLs can be scaled to high–dimensional regimes by tackling the optimal boundary control problem for a linear partial differential equation. In particular, we consider the *Timoshenko beam* (Macchelli and Melchiorri, 2004) model in Hamiltonian form. We derive formalize the boundary control problem and obtain a

### Neural Optimal Control via MSLs

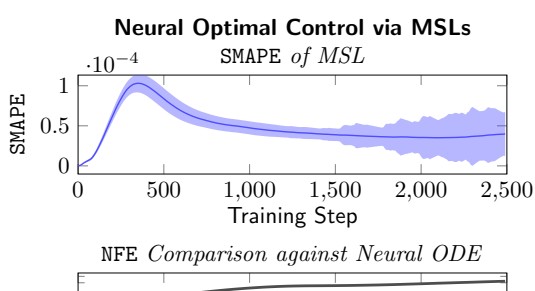

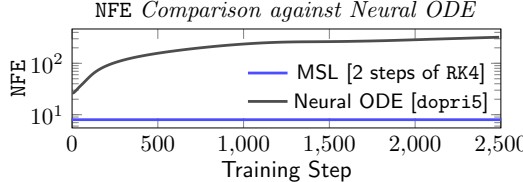

Figure 8: *Symmetric Mean Average Percentage Error* (SMAPE) between solutions of the controlled systems obtained by MSLs and nominal. Compared to Neural ODEs, MSLs solve the optimal control problem with NFE savings of several orders of magnitude by carrying forward their solution across training iterations.

structure–preserving spectral discretization yielding a 160–dimensional Hamiltonian ODE.

We parameterize the boundary control policy with a multi–layer perceptron taking as input (control feedback) the 160–dimensional discrete state. We train the model in similar setting to the previous example having the MSL equipped with `fw sensitivity` and one step of `rk4` for the parallel

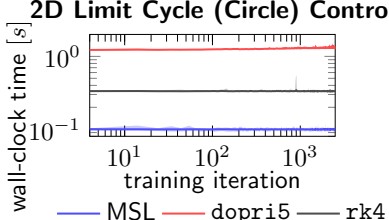

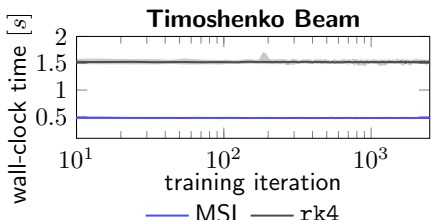

Figure 9: Mean and standard deviation of wall-clock time of complete training iteration (forward/backward passes + GD update) for different solvers on the *circle experiment*.

Figure 10: Mean and standard deviation of wall–clock time per training iteration for MSL and Neural ODE across training trials. MSLs require are three times faster than the sequential `rk4` with same accuracy (step size).

integration. We compare the training wall–clock time with a Neural ODE solved by sequential `rk4` in Fig. 10. The resulting speed up of MSL with forward sensitivity is three time faster than the baseline Neural ODE proving that the proposed method is able to scale to high–dimensional regimes. We include a formal treatment of the boundary control problem in Appendix D while further experimental details are provided in Appendix E.3.

## 4.3 Fast Neural CDEs for Time Series Classification

To further verify the broad range of applicability of MSLs, we apply them to time series classification as faster alternatives to *neural controlled differential equations* (Neural CDEs) (Kidger et al., 2020b). Here, MSLs remain applicable since Neural CDEs are practically solved as ODEs with a special structure, as described in Appendix E.4. We tackle the `PhysioNet` 2019 challenge (Goldberger et al., 2000) on `sepsis` prediction, following the exact experimental procedure described by Kidger et al. (2020b),

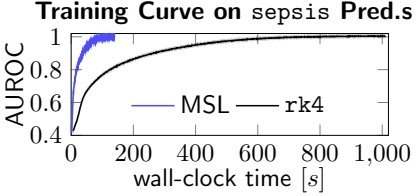

Figure 11: Mean and standard deviation of `AUROC` during training of MSLs and baseline Neural CDEs on sepsis prediction.

including hyperparameters and Neural CDE architectures. However, we train all models on the full dataset to enable application of the fixed point tracking technique for MSLs[4]. Figure 11 visualized training convergence of zeroth–order MSL Neural CDEs and the baseline Neural CDE solved with rk4 as in the original paper. Everything else being equal, including architecture and backpropagation via sequential adjoints, MSL Neural CDEs converge with total wall–clock time one order of magnitude smaller than the baseline.

## 5 Related Work

**Parallel–in–time integration & multiple shooting** MSLs belong to the framework of time–parallel integration algorithms. The study of these methods is relatively recent, with seminal work in the 60s (Nievergelt, 1964). The *multiple shooting* formulation of time–parallel integration, see e.g. (Bellen and Zennaro, 1989) or (Chartier and Philippe, 1993), finally lead to the modern algorithmic form using the Newton iteration reported in (3.2). *Parareal* (Lions et al., 2001) has been successively introduced as a cheaper approximated solution of multiple shooting problem, rapidly spreading across application domains, e.g. optimal control of partial differential equations (Maday and Turinici, 2002). We refer to (Gander, 2015, 2018) as excellent introduction to the topic. We also note recent work (Vialard et al., 2020) introducing *single shooting* terminology for Neural ODEs (Chen et al., 2018), albeit in the unrelated context of learning time–varying parameters.

---

[4]We note that the training for all models has been performed on a single NVIDIA RTX A6000 with 48Gb of GPU memory.

**Time–parallelization in neural models** In the pursuit for increased efficiency, several works have proposed approaches to parallelization across time or *depth* in neural networks. (Gunther et al., 2020; Kirby et al., 2020; Sun et al., 2020) use multigrid and penalty methods to achieve speedups in ResNets. Meng et al. (2020) proposed a parareal variant of Physics–informed neural networks (PINNs) for PDEs. Zhuang et al. (2021) uses a penalty–variant of multiple shooting with adjoint sensitivity for parameter estimation in the medical domain. Solving the boundary value problems with a regularization term, however, is not guaranteed to converge to a continuous solution. The method of Zhuang et al. (2021) further optimizes its parameters in a full–batch regime, where application of (1) achieves drastic speedups while preserving convergence guarantees. Recent theoretical work (Lorin, 2020) has applied *parareal* methods to Neural ODEs. However, their analysis is limited to the theoretical computational complexity setting and does not involve multiple shooting methods nor derives its implicit differentiation formula.

In contrast our objective is to introduce a novel class of implicit time–parallel models, and to validate their computational efficiency across settings.

# 6 Conclusion

This work introduces differentiable *Multiple Shooting Layers* (MSLs), a parallel–in–time alternative to neural differential equations. MSLs seek solutions of differential equations via parallel application of root finding methods across solution subintervals. Here, we analyze several model variants, further proving a fixed point tracking property that introduces drastic speedups in full–batch training. The proposed approach is validated on different tasks: as generative models, MSLs are shown to achieve same task performance as Neural ODE baselines with 60% less NFEs, whereas they are shown to offer several orders of magnitude faster in optimal control tasks.

*Remarkably few methods have been proposed for parallel integration of ODEs. In part this is because the problems do not have much natural parallelism. (Gear, 1988)*

# Funding Statement

This work was financially supported by The University of Tokyo, KAIST and RIKEN AIP. All experiments were run on GPUs provided by The University of Tokyo and KAIST.

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
