# Differentiable Multiple Shooting Layers
## *Supplementary Material*

## Table of Contents

# A Proofs

## A.1 Proof of Theorem 1

**Theorem 1** (Quadratic fixed-point tracking). *If $f_\theta$ is twice continuously differentiable in $z$ then*

$$\|B_{p+1}^* - \bar{B}_p^*\|_2 \le M\eta_p^2 \tag{3.4}$$

*for some $M > 0$. $\bar{B}_p^*$ is the result of one Newton iteration applied to $B_p^*$.*

*Proof.* For compactness, we neglect the dependence of $\gamma_\theta$, $g_\theta$ on $z_0$ and we write $\gamma(B, \theta) = \gamma_\theta(B, z_0)$, $g(B, \theta) = g_\theta(B, z_0)$. Recalling that by definition of $B_{p+1}^*$ it holds

$$g(B_{p+1}^*, \theta_{p+1}) = B_{p+1}^* - \gamma(B_{p+1}^*, \theta_{p+1}) = \mathbb{0}, \tag{A.1}$$

we write the 2-jet of the matching equation at $B_p^*$,

$$\begin{aligned}
g(B_{p+1}^*, \theta_{p+1}) = g(B_p^*, \theta_{p+1}) &+ \mathsf{D}g(B_p^*, \theta_{p+1})\left[B_{p+1}^* - B_p^*\right] \\
&+ \frac{1}{2}\mathsf{D}^2 g(B_p^*, \theta_{p+1})\left[B_{p+1}^* - B_p^*\right]^{\otimes 2} + o(\|B_{p+1}^* - B_p^*\|_2^3)
\end{aligned} \tag{A.2}$$

where $\mathsf{D}g$, $\mathsf{D}^2 g$ can be computed thanks to the assumptions on differentiability of $f_\theta$. From the Newton iteration we have that

$$\begin{aligned}
\bar{B}_p^* - B_p^* &= -[\mathsf{D}g(B_p^*, \theta_{p+1})]^{-1} g(B_p^*, \theta_{p+1}) \\
\Leftrightarrow \quad \mathbb{0} &= g(B_p^*, \theta_{p+1}) + \mathsf{D}g(B_p^*, \theta_{p+1})\left[\bar{B}_p^* - B_p^*\right]
\end{aligned} \tag{A.3}$$

Using $\mathbb{0} = g(B_{p+1}^*, \theta_{p+1})$, we subtract (A.3) from (A.2) yielding

$$\begin{aligned}
\mathbb{0} = &\cancel{g(B_p^*, \theta_{p+1})} + \mathsf{D}g(B_p^*, \theta_{p+1})\left[B_{p+1}^* - B_p^*\right] - \cancel{g(B_p^*, \theta_{p+1})} - \mathsf{D}g(B_p^*, \theta_{p+1})\left[\bar{B}_p^* - B_p^*\right] \\
&+ \frac{1}{2}\mathsf{D}^2 g(B_p^*, \theta_{p+1})\left[B_{p+1}^* - B_p^*\right]^{\otimes 2} + o(\|B_{p+1}^* - B_p^*\|_2^3) \\
= &\mathsf{D}g(B_p^*, \theta_{p+1})\left[B_{p+1}^* - B_p^*\right] - \mathsf{D}g(B_p^*, \theta_{p+1})\left[\bar{B}_p^* - B_p^*\right] \\
&+ \frac{1}{2}\mathsf{D}^2 g(B_p^*, \theta_{p+1})\left[B_{p+1}^* - B_p^*\right]^{\otimes 2} + o(\|B_{p+1}^* - B_p^*\|_2^3) \\
= &\mathsf{D}g(B_p^*, \theta_{p+1})\left[B_{p+1}^* - \bar{B}_p^*\right] + \frac{1}{2}\mathsf{D}^2 g(B_p^*, \theta_{p+1})\left[B_{p+1}^* - B_p^*\right]^{\otimes 2} + o(\|B_{p+1}^* - B_p^*\|_2^3).
\end{aligned} \tag{A.4}$$

Being the Jacobian of $g_\theta$

$$\mathsf{D}g(B_p^*, \theta_{p+1}) = \mathbb{I}_N \otimes \mathbb{I}_{n_z} - \mathsf{D}\gamma(B_p^*, \theta_{p+1}) = \begin{bmatrix} \mathbb{I}_{n_z} & \times & \times & \times \\ -\mathsf{D}\phi_{\theta_{p+1}, 0}(b_{0,p}^*) & \mathbb{I}_{n_z} & \times & \times \\ \times & \ddots & \ddots & \times \\ \times & \times & -\mathsf{D}\phi_{\theta_{p+1}, N-1}(b_{N-1,p}^*) & \mathbb{I}_{n_z} \end{bmatrix}$$

always invertible due to the nilpotency of $\mathsf{D}\gamma$, we can solve (A.4) in terms for $B_{p+1}^* - \bar{B}_p^*$, leading to

$$B_{p+1}^* - \bar{B}_p^* = \frac{1}{2}\left[\mathsf{D}g(B_p^*, \theta_{p+1})\right]^{-1} \mathsf{D}^2 g(B_p^*, \theta_{p+1})\left[B_{p+1}^* - B_p^*\right]^{\otimes 2} + o(\|B_{p+1}^* - B_p^*\|_2^3). \tag{A.5}$$

Taking the norm we have

$$\|B_{p+1}^* - \bar{B}_p^*\|_2 \le \frac{1}{2}\|[\mathsf{D}g(B_p^*, \theta_{p+1})]^{-1}\|_2 \|\mathsf{D}^2 g(B_p^*, \theta_{p+1})\|_2 \|B_{p+1}^* - B_p^*\|_2^2. \tag{A.6}$$

Using

$$\|B_{p+1}^* - B_p^*\|_2 \le \eta_p m_L^\theta m_\gamma^\theta$$

and $\|\mathsf{D}^2 g(B_p^*, \theta_{p+1})\|_2 = \|\mathsf{D}^2 \gamma(B_p^*, \theta_{p+1})\|_2 \le m_{\partial\gamma}^z$ we obtain

$$\|B_{p+1}^* - \bar{B}_p^*\|_2 \le \frac{1}{2}\eta_p^2 (m_L^\theta m_\gamma^\theta)^2 m_{\partial\gamma}^z \|[\mathsf{D}g(B_p^*, \theta_{p+1})]^{-1}\|_2. \tag{A.7}$$

Since $R = \mathsf{D}\gamma(B_p^*, \theta_{p+1})$ is a nilpotent matrix then

$$[\mathsf{D}g(B_p^*, \theta_{p+1})]^{-1} = [\mathbb{I}_N \otimes \mathbb{I}_{n_z} - R]^{-1} = \mathbb{I}_N \otimes \mathbb{I}_{n_z} + \sum_{n=1}^{N} R^n$$

and

$$\|[\mathsf{D}g(B_p^*, \theta_{p+1})]^{-1}\|_2 \leq 1 + \sum_{n=1}^{N} \|R\|_2^n.$$

By Lipsichitz continuity of $\gamma(B_p^*, \theta_{p+1})$ we have that

$$\|R\|_2 \leq m_\gamma^z$$

and

$$\|[\mathsf{D}g(B_p^*, \theta_{p+1})]^{-1}\|_2 \leq 1 + \sum_{n=1}^{N} (m_\gamma^z)^n$$

By convergence of finite geometric series we obtain

$$\|[\mathsf{D}g(B_p^*, \theta_{p+1})]^{-1}\|_2 \leq \frac{1 - (m_\gamma^z)^{N+1}}{1 - m_\gamma^z}$$

The final bound on the tracking error norm thus becomes

$$\|B_{p+1}^* - \bar{B}_p^*\|_2 \leq \frac{1}{2}\eta_p^2 (m_L^\theta m_\gamma^\theta)^2 m_{\partial\gamma}^z \frac{1 - (m_\gamma^z)^{N+1}}{1 - m_\gamma^z}. \tag{A.8}$$

The proof of the theorem is completed by setting

$$M > \frac{1}{2}(m_L^\theta m_\gamma^\theta)^2 m_{\partial\gamma}^z \frac{1 - (m_\gamma^z)^{N+1}}{1 - m_\gamma^z}$$

$\square$

## A.2 Proof of Proposition 1

**Proposition 1** (Forward Sensitivity (Khalil, 2002))**.** *Let $\phi_\theta(z, s, t)$ be the solution of* (2.1)*. Then,* $v(t) = \mathsf{D}\phi_\theta(z, s, t)$ *satisfies the linear matrix–valued differential equations*

$$\dot{v}(t) = \mathsf{D}f_\theta(t, z(t))v(t), \quad v(s) = \mathbb{I}_{n_z} \quad \text{where } \mathsf{D}f_\theta \text{ denotes } \partial f_\theta/\partial z.$$

*Proof.* The following proof is adapted from (Khalil, 2002, Section 3.3). If $\phi_\theta(z_0, s, t)$ is a solution of (2.1) at time $t$ starting from $z_0$ at time $s$, $s < t$; $s, t \in [t_0, t_N]$ then

$$\phi_\theta(z_0, s, t) = z(t) = z_0 + \int_s^t f_\theta(\tau, z(\tau))\mathrm{d}\tau \tag{A.9}$$

Differentiating under the integral sign w.r.t. $z$ yields

$$\begin{aligned}
\mathsf{D}\phi_\theta(z_0, s, t) = \frac{\mathrm{d}z(t)}{\mathrm{d}z_0} &= \frac{\mathrm{d}z_0}{\mathrm{d}z_0} + \int_s^t \frac{\partial f_\theta(\tau, z(\tau))}{\partial z(\tau)} \frac{\mathrm{d}z(\tau)}{\mathrm{d}z_0}\mathrm{d}\tau \\
&= \mathbb{I}_{n_z} + \int_s^t \frac{\partial f_\theta(\tau, z(\tau))}{\partial z(\tau)} \frac{\mathrm{d}z(\tau)}{\mathrm{d}z_0}\mathrm{d}\tau
\end{aligned} \tag{A.10}$$

We denote $\mathsf{D}f_\theta(t, z(t)) = \partial f_\theta(\tau, z(\tau))/\partial z(\tau)$ and we notice that $\mathrm{d}z(\tau)/\mathrm{d}z_0$ is the flow Jacobian $\mathsf{D}\phi_\theta(z_0, s, \tau)$ at time $\tau \in [s, t]$. Then, the function $v : [s, t] \to \mathbb{R}^{n_z \times n_z}; \ \tau \mapsto \mathsf{D}\phi_\theta(z_0, s, \tau)$ satisfies

$$v(t) = \mathbb{I}_{n_z} + \int_s^t \mathsf{D}f(\tau, z(\tau))v(\tau)\mathrm{d}\tau$$

or, in differential form, $v$ satisfies the IVP

$$\dot{v}(\tau) = \mathsf{D}f(\tau, z(\tau))v(\tau), \quad v(s) = \mathbb{I}_{n_z}.$$

$\square$

# B   Additional Theoretical Results

## B.1   Finite–Step Convergence

We discuss more rigorously the intuitions on the finite–step convergence of direct Newton methods introduced in the main text. The following results are thoroughly detailed in (Gander, 2018). We recall that, by assuming that the first shooting parameter is correctly initialized to $z_0$ and the numerical integration is *exact* (we can perfectly retrieve the sub–flows $\phi_{\theta,n}$), the shooting parameters $b_n^k$ coincides with the exact solution of (2.1) from Newton iteraion $k = n$ onward. Formally,

**Proposition 2** (Finite–step convergence). *If $b_0^0 = z_0$, then solution of the Newton iteration* (3.2) *are such that*

$$k \geq n \;\Rightarrow\; b_n^k = \phi_\theta(z_0, t_0, t_n). \tag{B.1}$$

*Proof.* The proof is obtained by induction on the shooting parameter index $n$ (*time* direction) and follows from (Gander, 2018, Theorem 2.3).

 i. (`base case: `$n = 0$) For $n = 0$, $b_0^0 = z_0$ by assumption. Moreover, the iteration (3.2) yields $b_0^k = b_0^{k+1} = z_0$ for all naturals $k$.

 ii. (`induction step: `$n \to n+1$) Suppose that

$$k \geq n \;\Rightarrow\; b_n^k = \phi_\theta(z_0, t_0, t_n).$$

We need to show that

$$k + 1 \geq n + 1 \;\Rightarrow\; b_{n+1}^{k+1} = \phi_\theta(z_0, t_0, t_{n+1}),$$

to conclude the proof by induction. We notice that if we increase $k$ to $k + 1$, then $k + 1$ is still greater than $n$ yielding $b_n^{k+1} = b_n^k = \phi_\theta(z_0, t_0, t_n)$. Using (3.2), we have

$$
\begin{aligned}
b_{n+1}^{k+1} &= \phi_{\theta,n}(b_n^k) + \mathsf{D}\phi_{\theta,n}(b_n^k)\left(b_n^{k+1} - b_n^k\right) \\
&= \phi_{\theta,n}(\phi_\theta(z_0, t_0, t_n)) + 0 && \text{by induction hypothesis } b_n^k = b_n^{k+1} = \phi_\theta(z_0, t_0, t_n); \\
&= \phi_\theta(z_0, t_0, t_{n+1}) && \text{by the flow property of ODE solutions;}
\end{aligned}
$$

where the induction hypothesis has been used thanks to the fact that $k + 1 \geq n + 1 \Rightarrow k \geq n$.

$\square$

The above result can be also extended to the zeroth–order (*parareal*) method as follows.

**Proposition 3** (Finite–step convergence w/ zeroth–order Jacobian approximation). *If $b_0^0 = z_0$, then solution of the approximate Newton iteration*

$$b_{n+1}^{k+1} = \phi_{\theta,n}(b_n^k) + \psi_{\theta,n}(b_n^{k+1}) - \psi_{\theta,n}(b_n^k) \tag{B.2}$$

*are such that*

$$k \geq n \;\Rightarrow\; b_n^k = \phi_\theta(z_0, t_0, t_n). \tag{B.3}$$

*Proof.* The proof is identical to the one Proposition 2 where (B.2) is used in the induction step and noticing that the correction term $\psi_{\theta,n}(b_n^{k+1}) - \psi_{\theta,n}(b_n^k)$ nullifies for $k > n$ by induction hypothesis $b_n^k = b_n^{k+1} = \phi_\theta(z_0, t_0, t_n)$. $\square$

Even though Proposition 2 and Proposition 3 show that the direct Newton method (and its zeroth-order approximation) will always converge to the exact solution of (2.1), full convergence after $N$ iterations is completely useless from a practical perspective. If we suppose to use a *fine solver* $\tilde{\phi}_{\theta,n}$ to obtain in parallel accurate numerical approximations of the sub–flows $\phi_{\theta,n}$ and we iterate (3.2) $N$ times, we will also have executed the parallel integration $N$ times. Thus, one could also just have applied the same fine solver sequentially across the $N$ boundary points $t_n$ with one processing thread and obtain the same result. For this reason, we believe that tracking Theorem 1 is a key result to obtain large speedups in the machine learning applications of MSLs.

## B.2 Flows Sensitivities $\mathsf{D}\phi_{\theta,n}$

The most computationally demanding stage of the MSL inference is without any doubts the correction term

$$\mathsf{D}\phi_{\theta,n}(b_n^k)\left(b_n^{k+1} - b_n^k\right)$$

of the direct Newton iteration (3.2). In this paper, we propose to either use the forward sensitivity approach of Proposition 1 or to rely on the zeroth–order approximation of parareal. Moreover, we discouraged the use of both reverse–mode AD and backward adjoint sensitivities to compute the full Jacobians $\mathsf{D}\phi_{\theta,n}$ due to their higher memory or computational cost.

**Sensitivities with** $N - k$ jvp**s** A common feature among the aforementioned approaches (but the parareal) is that all $\mathsf{D}\phi_{\theta,n}$ can be computed in parallel at the beginning of each Newton iteration with a single call of the sensitivity routine. An alternative *sequential* approach relies on computing $\mathsf{D}\phi_{\theta,n}(b_n^k)\left(b_n^{k+1} - b_n^k\right)$ directly as a jvp during each step of (3.2). This method avoids the computation of the full Jacobians at cost of having to call the jvp routine $N - k$ times at each Newton iteration. In such case the only parallel operation performed is the integration of the sub–flows $\phi_{\theta,n}$. Nonetheless, we believe that this direction is worth to be explored in future works.

## B.3 Backward Model of Multiple Shooting Layers

We show how MSLs can be trained via standard gradient descent techniques where gradients can be either computed by back–propagating through the operations of the forward pass (parallel/memory intensive) or by using the convergence property of direct Newton method and directly apply a interpolated adjoint routine[5] (sequential/memory efficient). Although we believe that these two approaches to backpropagation are sufficient within the scope of this manuscript as they allow for substantial computational speedups and robustness, we hereby report further theoretical considerations on the backward pass of MSLs. A thorough algorithmic and experimental analysis of the following content is a promising research direction for future work.

**Implicit differentiation of MSLs** As repeatedly pointed out throughout the paper, MSLs are implicit models and satisfy the implicit relation

$$B^* \; : \; B^* = \gamma_\theta(B^*, z_0). \tag{B.4}$$

It thus make sense to interpret the backward pass of MSLs in an *implicit* sense. In particular, implicit differentiation of the relation (B.4) at $B^*$ leads to the following loss gradients.

**Theorem 2** (Implicit Gradients). *Consider a smooth loss function $L_\theta$. It holds*

$$\frac{\mathrm{d}L_\theta}{\mathrm{d}\theta} = \frac{\partial L_\theta}{\partial \theta} + \frac{\partial L_\theta}{\partial \ell_y}\frac{\partial \ell_y}{\partial B^*}\left[\mathbb{I}_{n_z} \otimes \mathbb{I}_N - \mathsf{D}\gamma_\theta(B^*)\right]^{-1}\frac{\partial \gamma_\theta}{\partial \theta} \tag{B.5}$$

*where $\mathsf{D}\gamma_\theta(B^*, z_0) \in \mathbb{R}^{Nn_z \times Nn_z}$ is the Jacobian of $\gamma_\theta$ computed at $B^*$.*

*Proof.* By application of the chain rule to the MSL forward model (2.3) we obtain

$$\frac{\mathrm{d}L_\theta}{\mathrm{d}\theta} = \frac{\partial L_\theta}{\partial \theta} + \frac{\partial L_\theta}{\partial \ell_y}\frac{\partial \ell_y}{\partial B^*}\frac{\mathrm{d}B^*}{\mathrm{d}\theta}.$$

With

$$g_\theta(B^*) = \mathbb{0} \;\Rightarrow\; B^* - \gamma(B^*, z_0) = \mathbb{0}$$

$$\Leftrightarrow \frac{\partial g_\theta(B^*)}{\partial \theta} + \left[\mathbb{I}_{n_z} \otimes \mathbb{I}_N - \mathsf{D}\gamma_\theta(B^*)\right]\frac{\mathrm{d}B^*}{\mathrm{d}\theta} = 0$$

$$\Leftrightarrow \frac{\mathrm{d}B^*}{\mathrm{d}\theta} = \left[\mathbb{I}_{n_z} \otimes \mathbb{I}_N - \mathsf{D}\gamma_\theta(B^*)\right]^{-1}\frac{\partial \gamma_\theta}{\partial \theta}.$$

Thus,

$$\frac{\mathrm{d}L_\theta}{\mathrm{d}\theta} = \frac{\partial L_\theta}{\partial \theta}\frac{\partial L_\theta}{\partial \ell_y}\frac{\partial \ell_y}{\partial B^*}\left[\mathbb{I}_{n_z} \otimes \mathbb{I}_N - \mathsf{D}\gamma_\theta(B^*)\right]^{-1}\frac{\partial \gamma_\theta}{\partial \theta}$$

---

[5]Implementation details are provided in Appendix C.4

where the Jacobian $\mathsf{D}\gamma_\theta(B^*)$ is computed as

$$\mathsf{D}\gamma_\theta(B^*) = \begin{bmatrix} \times & \times & \times & \times & \times \\ \mathsf{D}\phi_{\theta,0}(b_0^*) & \times & \times & \times & \times \\ \times & \mathsf{D}\phi_{\theta,1}(b_1^*) & \times & \times & \times \\ \times & \times & \ddots & \ddots & \times \\ \times & \times & \times & \mathsf{D}\phi_{\theta,N-1}(b_{N-1}^*) & \times \end{bmatrix}$$

and

$$\frac{\partial\gamma_\theta}{\partial\theta} = \begin{bmatrix} \mathbb{0}_{n_z} \\ \dfrac{\partial\phi_{\theta,0}(b_0)}{\partial\theta} \\ \vdots \\ \dfrac{\partial\phi_{\theta,N-1}(b_{N-1})}{\partial\theta} \end{bmatrix}$$

$\square$

The implicit differentiation routine suggested by Theorem 2 presents two terms which appear to be very demanding both memory and computation–wise:

$(i)$ The inverse Jacobian $[\mathbb{1}_{n_z} \otimes \mathbb{1}_N - \mathsf{D}\gamma_\theta(B^*)]^{-1}$ of the implicit relation;

$(ii)$ The sub–flows sensitivities to the model parameters $\theta$.

In order to retrieve $(i)$ in standard *Deep Equilibrium Models* Bai et al. (2019), one should either compute the full–Jacobian at the fixed point via AD and invert it or "recycle" its low–rank approximation from the Quasi–Newton method of the forward pass. In the case of MSLs we can take advantage of the special structure of the implicit relation to obtain the exact Jacobian inverse in a computationally efficient manner. In fact, if `fw-sensitivity` has been used in the forward pass to compute $B^*$, then the sensitivities of the sub–flows computed at the last step $K$ of the Newton iteration $\mathsf{D}\phi_{\theta,n}(b_n^K)$ can be stored and re–used to construct the Jacobian $\mathsf{D}\gamma_\theta$. Further, due to the nilpotency of $\mathsf{D}\gamma_\theta$ the inverse of the total Jacobian can be retrieved in closed form by the finite matrix power series

$$[\mathbb{1}_{n_z} \otimes \mathbb{1}_N - \mathsf{D}\gamma_\theta(B^*)]^{-1} = \mathbb{1}_{n_z} \otimes \mathbb{1}_N + \sum_{n=1}^{N} [\mathsf{D}\gamma_\theta(B^*)]^n.$$

Finally, $(ii)$ may be indirectly computed with a single `vjp`

$$v^\top \frac{\partial\gamma_\theta(B^*, z_0)}{\partial\theta}$$

with $v^\top$ being a 1 by $Nn_\theta$ row vector defined as

$$v^\top = \frac{\partial L_\theta}{\partial\ell_y} \frac{\partial\ell_y}{\partial B^*} \left[ \mathbb{1}_{n_z} \otimes \mathbb{1}_N + \sum_{n=1}^{N} [\mathsf{D}\gamma_\theta(B^*)]^n \right]$$

leading to the implicit cost gradient with a single call of the AD.

# C  Additional Details on the Realization of MSLs

Effective time–parallelization of MSLs requires implementation of specialized computational primitives. In example, forward sensitivity methods benefit from a breakdown of matrix–Jacobian products into a vmapped vector–Jacobian products. Here, we provide code for several key methods and classes which have been incorporated in the `torchdyn` (Poli et al., 2020b) library for neural differential equations and implicit models.

## C.1  Software Implementation of Forward Sensitivity

Forward sensitivity analysis is extensively used in MSLs to compute $\mathrm{d}\phi_{\theta,n}/\mathrm{d}b_n$ in parallel for each shooting parameter $b_n,\ n = 0, \dots, N-1$. We showcase how this can be efficiently implemented in `Pytorch` (Paszke et al., 2019). Although the implementation fully accommodates batches $n_b$ of data, i.e. each $b_n$ is a $n_b$ by $n_z$ matrix, we will limit the algorithmic analysis to the unitary batch dimension. The forward sensitivity algorithm aims at computing the solution of the differential equation

$$\begin{pmatrix} \dot{z}_n(t) \\ \dot{v}_n(t) \end{pmatrix} = \begin{pmatrix} f_\theta(t, z_n(t)) \\ \mathsf{D}f_\theta(t, z_n(t))v_n(t) \end{pmatrix}, \quad \begin{pmatrix} \dot{z}_n(0) \\ \dot{v}_n(0) \end{pmatrix} = \begin{pmatrix} b_n \\ \mathbb{I}_{n_z} \end{pmatrix}, \quad t \in [t_n, t_{n+1}] = \mathcal{T}_n$$

for all $n$, to return $\phi_{\theta,n} = z_n(t_{n+1})$ and $\mathrm{d}\phi_{\theta,n}/\mathrm{d}b_n = v_n(t_{n+1})$. Let $Z$ and $V$ be the tuples containing all $z_n$ and $v_n$,

$$Z = (z_0, z_1, \dots, z_{N-1}) \in \overbrace{\mathbb{R}^{n_z} \times \cdots \times \mathbb{R}^{n_z}}^{N} \equiv \mathbb{R}^{N \times n_z}$$

$$V = (v_0, v_1, \cdots, v_{N-1}) \in \underbrace{\mathbb{R}^{n_z \times n_z} \times \cdots \times \mathbb{R}^{n_z \times n_z}}_{N} \equiv \mathbb{R}^{N \times n_z \times n_z}.$$

Given a tuple of time instants $T = (\tau^0, \tau^1, \dots, \tau^{N-1}) \in [t_0, t_1] \times [t_1, t_1] \times [t_{N-1}, t_N] \subset \mathbb{R}^{N \times 1}$, $f_\theta$ can evaluated in parallel on $T$ and $Z$ as the number $N$ of shooting parameters $b_n$ and sub–intervals $[t_n, t_{n+1}]$ only accounts for a *batch* dimension. From a software perspective, we can obtain

$$F(T, Z) = (f_\theta(\tau^0, z_0), \dots, f_\theta(\tau^{N-1}, z_{N-1}))$$

in a single call of the function $f_\theta$, e.g. an instantiated `PyTorch`'s `nn.Module` object. Conversely, when attempting to compute "$\frac{\partial F}{\partial Z}V$" in parallel, additional software infrastructure is necessary. The main obstacle is that each Jacobian–matrix product (`jmp`)

$$\frac{\partial f_\theta(t, z_n(t))}{\partial z_n} v_n(t)$$

generally requires $n_z$ autograd calls. Following the `Jax`'s (Bradbury et al., 2018) approach, we make use of a `PyTorch` implementation[6] of *vectorizing maps* (`vmaps`) to distribute the computation of the individual Jacobian–vectors products (in *batch* for each $n = 0, \dots, N-1$) and compose the `jmp` row–by–row or column–by–column. In particular we define the `vmapped_jmp` function

```
1    def vmapped_jmp(y, z, v):
2        """ Parallel computation of matrix Jacobian products with vmap
3        """
4        def get_jvp(v):
5            return torch.autograd.grad(y, z, v, retain_graph=True)[0]
6        return vmap(get_jvp, in_dims=2, out_dims=2)(v)
```

The forward sensitivity can then be computed as follows

```
1    class ForwardSensitivity(nn.Module):
2        "Forward sensitivity for ODEs. Integrates the ODE returning the state and
     ↪   forward sensitivity"
3        def __init__(self, f):
4            super().__init__()
```

---

[6]see https://pytorch.org/docs/master/generated/torch.vmap.html

```
5            self.f = f
6
7        def forward(self, z0, t_span, odeint_func, solver='rk4', atol=1e-5,
         ↪  rtol=1e-5):
8            I = eye(z0.shape[-1]).to(z0)
9            # handle regular `batch, dim` case as well as `seq_dim, batch, dim`
10           v0 = I.repeat(z0.shape[0], 1, 1) if len(z0.shape) < 3 else
         ↪  I.repeat(*z0.shape[:2], 1, 1)
11
12           self.z_shape, self.v_shape = z0.shape, v0.shape
13
14           zv0 = self._ravel_state(z0, v0)
15           zvT = odeint_func(self._sensitivity_dynamics, zv0, t_span,
16                       solver=solver)[1]
17           zT, vT = self._unravel_state(zvT)
18           return zT, vT
19
20       def _sensitivity_dynamics(self, t, zv):
21           z, v = self._unravel_state(zv)
22           # compute vector field
23           dz = self.f(t, z.requires_grad_(True))
24           # compute fw sensitivity via mjp
25           dv = self._mjp(dz, z, v)
26           return self._ravel_state(dz, dv)
27
28       def _jmp(self, f, z, v):
29           """Parallel computation of matrix jacobian products with vmap
30           """
31           def get_vjp(v):
32               return torch.autograd.grad(f, z, v, retain_graph=True)[0]
33           return vmap(get_vjp, in_dims=2, out_dims=2)(v)
34
35       def _ravel_state(self, z, v):
36           v = v.reshape(*z.shape[:-1], -1)
37           zv = torch.cat([z, v], -1)
38           return zv
39
40       def _unravel_state(self, zv):
41           z, v = zv[...,:self.z_shape[-1]], zv[...,self.z_shape[-1]:]
42           v = v.reshape(*z.shape, self.z_shape[-1])
43           return z, v
```

where `odeint` is ODE solver utility of the `torchdyn` (Poli et al., 2020b) library.

## C.2 Implementation of Direct Newton Method

*Forward sensitivity Newton* (`fw sensitivity`) MSL is a variant of the proposed model class which obtains the quantities $D\phi_{\theta,n}$ directly by augmenting the time–parallelized forward dynamics through the `ForwardSensitivity` class previously detailed. During the evaluation of the *advancement* function $\gamma_\theta(B, z_0) = (z_0, \phi_\theta(b_0, t_0, t_1), \ldots, \phi_\theta(b_{N-1}, t_{N-1}, t_N))$, `ForwardSensitivity` maximizes reutilization of vector field $f_\theta$ evaluations by leveraging the results to advance both *standard* as well as sensitivity dynamics. This provides an overall reduction in the potentially expensive evaluation of the neural network $f_\theta$, compared to *parareal* (zeroth–order MSL). We hereby report the `PyTorch` implementation for both the `fw sensitivity` MSL and zeroth–order MSL methods

```
1  class MSForward(MShootingSolverTemplate):
2      """Multiple shooting solver using forward sensitivity analysis on the
       ↪  matching conditions of shooting parameters"""
3      def __init__(self, coarse_method='euler', fine_method='rk4'):
4          super().__init__(coarse_method, fine_method)
5          self.fsens = None
6
```

```
7       def root_solve(self, odeint_func, f, x, t_span, B, fine_steps, maxiter):
8           if self.fsens is None:
9               self.fsens = ForwardSensitivity(f)
10
11          dt, n_subinterv = t_span[1] - t_span[0], len(t_span)
12          sub_t_span = torch.linspace(0, dt, fine_steps).to(x)
13          i = 0
14          while i <= maxiter:
15              i += 1
16              with torch.set_grad_enabled(True):
17                  B_fine, V_fine = self.fsens(B[i-1:], sub_t_span,
                        ↪  odeint_func=odeint_func,
18                                               solver=self.fine_method)
19              B_fine, V_fine = B_fine[-1], V_fine[-1]
20              B_out = torch.zeros_like(B)
21              B_out[:i] = B[:i]
22              B_in = B[i-1]
23              for m in range(i, n_subinterv):
24                  B_in = B_fine[m-i] + torch.einsum('bij, bj -> bi', V_fine[m-i],
                        ↪  B_in - B[m-1])
25                  B_out[m] = B_in
26              B = B_out
27          return B
```

```
1   class MSZero(MShootingSolverTemplate):
2       def __init__(self, coarse_method='euler', fine_method='rk4'):
3           """Multiple shooting solver using Parareal updates (zero-order
                ↪  approximation of the Jacobian)
4           """
5           super().__init__(coarse_method, fine_method)
6       def root_solve(self, odeint_func, f, x, t_span, B, fine_steps, maxiter):
7           dt, n_subinterv = t_span[1] - t_span[0], len(t_span)
8           sub_t_span = torch.linspace(0, dt, fine_steps).to(x)
9           i = 0
10          while i <= maxiter:
11              i += 1
12              B_coarse = odeint_func(f, B[i-1:], sub_t_span,
                    ↪  solver=self.coarse_method)[1][-1]
13              B_fine = odeint_func(f, B[i-1:], sub_t_span,
                    ↪  solver=self.fine_method)[1][-1]
14              B_out = torch.zeros_like(B)
15              B_out[:i] = B[:i]
16              B_in = B[i-1]
17              for m in range(i, n_subinterv):
18                  B_in = odeint_func(f, B_in, sub_t_span,
                        ↪  solver=self.coarse_method)[1][-1]
19                  B_in = B_in - B_coarse[m-i] + B_fine[m-i]
20                  B_out[m] = B_in
21              B = B_out
22          return B
```

In the above, we employ the finite–step convergence property of Newton MSL iterations to avoid redundant computation. More specifically, at iteration $k$ we do not advance shooting parameters $b_n, \; n < k$ by slicing the tensor $B$ during $\gamma_\theta$ evaluations. Similarly, updates in the form (3.2) are not performed for shooting parameters already at convergence.

### C.3    Alternative Approaches to MSL Inference

**On Newton and Quasi-Newton methods for MSL**    The root–finding problem arising in MSLs can also be approached by standard application of Newton or Quasi–Newton algorithms. Although Quasi–Newton algorithms can provide improved computational efficiency by maintaining a low–rank approximation of the Jacobian $Dg_\theta(B, z_0)$ rather than computing it from scratch every iteration, this

advantage does not translate well to the MSL case. Popular examples include, e.g., the Broyden family Broyden (1965) employed in *Deep Equilibrium Models* (DEQs) Bai et al. (2019). As discussed in the main text, thanks to the special structure of the Jacobian of the MSL problem, the direct Newton algorithm (3.2) can be applied without computation and inversion of the full Jacobian.

**Root finding via gradient descent**    A completely different approach to solve the implicit forward MSL pass (2.3) is to tackle the root–finding via some gradient–descent (GD) method minimizing $\|g_\theta(B)\|_2^2$, i.e.

$$B^* = \operatorname{argmin} \frac{1}{2}\|g_\theta(B)\|_2^2.$$

In the case of MSL, all GD solutions (i.e. minima of $\|g_\theta(B)\|_2^2$) are the same of the the root finding ones. This can be intuitively checked by inspecting the zeros of the gradient, i.e.

$$\nabla_B \frac{1}{2}\|g_\theta(B)\|_2^2 = Dg_\theta|_B g_\theta(B)$$

and, since $Dg_\theta|_B$ is nonsingular for all $B$,

$$\forall \tilde{B}^* \ : \ \nabla_B \frac{1}{2}\|g_\theta(\tilde{B}^*)\|_2^2 = \mathbb{0} \Rightarrow g_\theta(\tilde{B}^*) = \mathbb{0}.$$

### C.4   Implementation of Backward Interpolated Adjoint

We provide pseudo–code for our implementation of MSLs with backward gradients obtained via interpolated adjoints. The implementation relies on cubic interpolation utilities provided by `torchcde` Kidger et al. (2020b). Interpolation is used to obtain values of $z(t)$ without a full backsolve from $z(T)$.

```python
def _gather_odefunc_interp_adjoint(vf, vf_params, solver, atol, rtol,
    interpolator, solver_adjoint, atol_adjoint, rtol_adjoint, integral_loss,
    problem_type, maxiter=4, fine_steps=4):
    "Prepares definition of autograd.Function for interpolated adjoint
        sensitivity analysis of the above `ODEProblem`"
    class _ODEProblemFunc(Function):
        @staticmethod
        def forward(ctx, vf_params, x, t_span, B=None):
            t_sol, sol = generic_odeint(problem_type, vf, x, t_span, solver,
                atol, rtol, interpolator, B,
                                        True, maxiter, fine_steps)
            ctx.save_for_backward(sol, t_span, t_sol)
            return t_sol, sol

        @staticmethod
        def backward(ctx, *grad_output):
            sol, t_span, t_sol = ctx.saved_tensors
            vf_params = torch.cat([p.contiguous().flatten() for p in
                vf.parameters()])

            # initialize adjoint state
            xT, λT, µT = sol[-1], grad_output[-1][-1],
                torch.zeros_like(vf_params)
            λT_nel, µT_nel = λT.numel(), µT.numel()
            xT_shape, λT_shape, µT_shape = xT.shape, λT.shape, µT.shape
            A = torch.cat([λT.flatten(), µT.flatten()])

            spline_coeffs = natural_cubic_coeffs(x=sol.permute(1, 0,
                2).detach(), t=t_sol)
            x_spline = CubicSpline(coeffs=spline_coeffs, t=t_sol)

            # define adjoint dynamics
            def adjoint_dynamics(t, A):
                if len(t.shape) > 0: t = t[0]
                x = x_spline.evaluate(t).requires_grad_(True)
```

```python
29              t = t.requires_grad_(True)
30              λ, μ = A[:λT_nel], A[-μT_nel:]
31              λ, μ = λ.reshape(λT.shape), μ.reshape(μT.shape)
32              with torch.set_grad_enabled(True):
33                  dx = vf(t, x)
34                  dλ, dt, *dμ = tuple(grad(dx, (x, t) +
                    ↪ tuple(vf.parameters()), -λ,
35                                      allow_unused=True,
                                       ↪ retain_graph=False))
36
37                  if integral_loss:
38                      dg = torch.autograd.grad(integral_loss(t, x).sum(), x,
                        ↪ allow_unused=True, retain_graph=True)[0]
39                      dλ = dλ - dg
40
41                  dμ = torch.cat([el.flatten() if el is not None else
                    ↪ torch.zeros(1)
42                                  for el in dμ], dim=-1)
43              return torch.cat([dλ.flatten(), dμ.flatten()])
44
45          # solve the adjoint equation
46          n_elements = (λT_nel, μT_nel)
47          for i in range(len(t_span) - 1, 0, -1):
48              t_adj_sol, A = odeint(adjoint_dynamics, A, t_span[i - 1:i +
                ↪ 1].flip(0), solver, atol=atol, rtol=rtol)
49              # prepare adjoint state for next interval
50              A = torch.cat([A[-1, :λT_nel], A[-1, -μT_nel:]])
51              A[:λT_nel] += grad_output[-1][i - 1].flatten()
52
53          λ, μ = A[:λT_nel], A[-μT_nel:]
54          λ, μ = λ.reshape(λT.shape), μ.reshape(μT.shape)
55          return (μ, λ, None, None, None)
56
57      return _ODEProblemFunc
```

### C.5 Broader Impact

Differential equations are the language of science and engineering. As methods (Jia and Benson, 2019) and software frameworks (Rackauckas et al., 2019; Li et al., 2020; Poli et al., 2020b) are improved, yielding performance gains or speedups (Poli et al., 2020a; Kidger et al., 2020a; Pal et al., 2021), the range of applicability of neural differential equations is extended to more complex and larger scale problems. As with other techniques designed to reduce overall training time, we expect a net positive environment impact from the adoption of MSLs in the framework.

Application domains for MSLs include environments with real–time constraints, for example control and high frequency time series prediction. Shorter inference wall–clock and training iteration times should yield more robust models that can, in example, be retrained online at higher frequencies as more data is collected.

# D    Neural Network Control of the Timoshenko Beam

In this section we derive the dynamic model of the Timoshenko beam, the boundary control and the structure–preserving discretization of the problem.

## D.1    Port–Based Modeling of the Timoshenko Beam

Linear distributed port-Hamiltonian systems (Macchelli et al., 2004) in one-dimensional domains take the form

$$\frac{\partial z}{\partial t}(x,t) = P_1 \frac{\partial}{\partial x}(\mathcal{L}(x)z(x,t)) + (P_0 - G_0)\mathcal{L}(x)z(x,t) \tag{D.1}$$

with distributed state $z \in \mathbb{R}^{n_z}$ and spatial variable $x \in [a,b]$. Moreover, $P_1 = P_1^\top$ and invertible, $P_0 = -P_0^\top$, $G_0 = G_0^\top \geq 0$, and $\mathcal{L}(\cdot)$ is a bounded and Lipschitz continuous matrix-valued function such that $\mathcal{L}(x) = \mathcal{L}^\top(x)$ and $\mathcal{L}(x) \geq \kappa I$, with $\kappa > 0$, $\forall x \in [a,b]$. Given the Hamiltonian (total energy) of the system

$$H = \|z\|_{\mathcal{L}}^2 = \langle z,\ \mathcal{L}z \rangle_{L^2},$$

its variational derivative corresponds to the term $\mathcal{L}(x)z(x,t)$:

$$\frac{\delta H}{\delta z}(z(x,t),x) = \mathcal{L}(x)z(x,t)$$

A particular example from continuum mechanics that falls within the systems class (D.1) is the Timoshenko beam with no dissipation (Macchelli and Melchiorri, 2004). This system takes the following form:

$$\frac{\partial}{\partial t}\begin{pmatrix} p_t \\ p_r \\ \varepsilon_r \\ \varepsilon_t \end{pmatrix} = \begin{bmatrix} 0 & 0 & 0 & \partial_z \\ 0 & 0 & \partial_z & 1 \\ 0 & \partial_z & 0 & 0 \\ \partial_z & -1 & 0 & 0 \end{bmatrix} \left( \begin{bmatrix} (\rho A)^{-1} & 0 & 0 & 0 \\ 0 & (I_\rho)^{-1} & 0 & 0 \\ 0 & 0 & EI & 0 \\ 0 & 0 & 0 & K_{\mathrm{sh}}GA \end{bmatrix} \begin{pmatrix} p_t \\ p_r \\ \varepsilon_r \\ \varepsilon_t \end{pmatrix} \right), \tag{D.2}$$

where $\rho$ is the mass density, $A$ is the cross section area, $I_\rho$ is the rotational inertia, $E$ is the Young modulus, $I$ the cross section moment of area, $K_{\mathrm{sh}} = 5/6$ is the shear correction factor and $G$ the shear modulus.

For this examples the matrices $P_0,\ G_0,\ P_1,\ \mathcal{L}$ and are given by

$$P_0 = \begin{bmatrix} 0 & 0 & 0 & 0 \\ 0 & 0 & 0 & 1 \\ 0 & 0 & 0 & 0 \\ 0 & -1 & 0 & 0 \end{bmatrix}, \quad P_1 = \begin{bmatrix} 0 & 0 & 0 & 1 \\ 0 & 0 & 1 & 0 \\ 0 & 1 & 0 & 0 \\ 1 & 0 & 0 & 0 \end{bmatrix}, \quad G_0 = \mathbb{0}_{4\times4},$$

$$\mathcal{L}(z) = \begin{bmatrix} (\rho A)^{-1} & 0 & 0 & 0 \\ 0 & (I_\rho)^{-1} & 0 & 0 \\ 0 & 0 & EI & 0 \\ 0 & 0 & 0 & K_{\mathrm{sh}}GA \end{bmatrix}. \tag{D.3}$$

We investigate the boundary control of the Timoshenko beam model. As control input, the following selection is made (cantilever-free beam)

$$\pi_\partial = \begin{pmatrix} EI\varepsilon_r(b,t) \\ K_{\mathrm{sh}}GA\varepsilon_t(b,t) \\ (\rho A)^{-1}p_t(a,t) \\ (I_\rho)^{-1}p_r(a,t) \end{pmatrix} \tag{D.4}$$

Notice that the control expression can be rewritten compactly as follows

$$\pi_\partial = \mathcal{B}_\partial \begin{pmatrix} \mathcal{L}x(b,t) \\ \mathcal{L}x(a,t) \end{pmatrix}, \quad \text{where} \quad \mathcal{B}_\partial = \begin{bmatrix} 0 & 0 & 1 & 0 & 0 & 0 & 0 & 0 \\ 0 & 0 & 0 & 1 & 0 & 0 & 0 & 0 \\ 0 & 0 & 0 & 0 & 1 & 0 & 0 & 0 \\ 0 & 0 & 0 & 0 & 0 & 1 & 0 & 0 \end{bmatrix}. \tag{D.5}$$

To put system (D.2) in impedance form, the outputs are selected as follows

$$
y_\partial = \begin{pmatrix} (\rho A)^{-1} p_t(b,t) \\ (I_\rho)^{-1} p_r(b,t) \\ -EI\varepsilon_r(a,t) \\ -K_{\mathrm{sh}}GA\varepsilon_t(a,t) \end{pmatrix} \tag{D.6}
$$

This is compactly written as

$$
y_\partial = \mathcal{C}_\partial \begin{pmatrix} \mathcal{L}x(b,t) \\ \mathcal{L}x(a,t) \end{pmatrix}, \quad \text{where} \quad \mathcal{C}_\partial = \begin{bmatrix} 1 & 0 & 0 & 0 & 0 & 0 & 0 & 0 \\ 0 & 1 & 0 & 0 & 0 & 0 & 0 & 0 \\ 0 & 0 & 0 & 0 & 0 & 0 & -1 & 0 \\ 0 & 0 & 0 & 0 & 0 & 0 & 0 & -1 \end{bmatrix}. \tag{D.7}
$$

With this selection of inputs and outputs, the rate of the Hamiltonian is readily computed

$$
\dot{H} = \pi_\partial^\top y_\partial. \tag{D.8}
$$

Within the purpose of this paper we restrict to the case of *a cantilever beam undergoing a control action at the free end*

$$
\pi_\partial = \begin{pmatrix} \pi_{\partial,1} \\ \pi_{\partial,2} \\ 0 \\ 0 \end{pmatrix}, \tag{D.9}
$$

where $\pi_{\partial,1}$ is the control torque and $u_{\partial,2}$ is the control force.

## D.2 Discretization of the Problem

To discretize system (D.2), since the problem is linear, one can either rely on a energy formulation or a co-energy one. Given the coenergy variables

$$
\begin{pmatrix} v_t \\ v_r \\ \sigma_r \\ \sigma_t \end{pmatrix} = \begin{bmatrix} (\rho A)^{-1} & 0 & 0 & 0 \\ 0 & I_\rho^{-1} & 0 & 0 \\ 0 & 0 & EI & 0 \\ 0 & 0 & 0 & K_{\mathrm{sh}}GA \end{bmatrix} \begin{pmatrix} p_t \\ p_r \\ \varepsilon_r \\ \varepsilon_t \end{pmatrix} \tag{D.10}
$$

and introducing the bending and shear compliance

$$
C_b = (EI)^{-1}, \qquad C_s = (K_{\mathrm{sh}}GA)^{-1}, \tag{D.11}
$$

system (D.2) is rewritten as

$$
\begin{bmatrix} \rho A & 0 & 0 & 0 \\ 0 & I_\rho & 0 & 0 \\ 0 & 0 & C_b & 0 \\ 0 & 0 & 0 & C_s \end{bmatrix} \frac{\partial}{\partial t} \begin{pmatrix} v_t \\ v_r \\ \sigma_r \\ \sigma_t \end{pmatrix} = \begin{bmatrix} 0 & 0 & 0 & \partial_z \\ 0 & 0 & \partial_z & 1 \\ 0 & \partial_z & 0 & 0 \\ \partial_z & -1 & 0 & 0 \end{bmatrix} \begin{pmatrix} v_t \\ v_r \\ \sigma_r \\ \sigma_t \end{pmatrix}, \tag{D.12}
$$

A weak form suitable for mixed finite elements is readily obtained by considering its weak form using test functions $(\mu_t, \mu_r, \nu_r, \nu_t)$ and the integration by parts applied to the first two lines. In this formulation, the Dirichlet boundary condition have to be incorporated as essential boundary conditions

$$
\begin{aligned}
\langle \mu_t, \rho A \partial_t v_t \rangle_\Omega &= - \langle \partial_z \mu_t, \sigma_t \rangle_\Omega + \mu_t(b)\pi_{\partial,2}, \\
\langle \mu_r, I_\rho \partial_t v_r \rangle_\Omega &= - \langle \partial_z \mu_r, \sigma_r \rangle_\Omega + \langle \mu_r, \sigma_t \rangle_\Omega + \mu_r(b)\pi_{\partial,1}, \\
\langle \nu_r, C_b \partial_t \sigma_r \rangle_\Omega &= \langle \nu_r, \partial_z v_r \rangle_\Omega, \\
\langle \nu_t, C_s \partial_t \sigma_t \rangle_\Omega &= \langle \nu_t, \partial_z v_t \rangle_\Omega - \langle \nu_t, v_r \rangle_\Omega,
\end{aligned} \tag{D.13}
$$

where $\Omega = [a,b]$ and $\langle f, g \rangle_\Omega = \int_a^b fg \, \mathrm{d}x$. Introducing the following Galerkin basis functions

$$
\begin{aligned}
\mu_t &= \sum_{i=1}^{N_{v_t}} \varphi_{v_t}^i \mu_t^i, & \mu_r &= \sum_{i=1}^{N_{v_r}} \varphi_{v_r}^i \mu_r^i, & \nu_r &= \sum_{i=1}^{N_{\sigma_r}} \varphi_{\sigma_r}^i \nu_r^i, & \nu_t &= \sum_{i=1}^{N_{\sigma_t}} \varphi_{\sigma_t}^i \nu_r^i, \\
v_t &= \sum_{i=1}^{N_{v_t}} \varphi_{v_t}^i v_t^i, & v_r &= \sum_{i=1}^{N_{v_r}} \varphi_{v_r}^i v_r^i, & \sigma_r &= \sum_{i=1}^{N_{\sigma_r}} \varphi_{\sigma_r}^i \sigma_r^i, & \sigma_t &= \sum_{i=1}^{N_{\sigma_t}} \varphi_{\sigma_t}^i \sigma_r^i,
\end{aligned} \tag{D.14}
$$

a finite-dimensional system is obtained

$$
\begin{bmatrix} M_{\rho A} & \times & \times & \times \\ \times & M_{I_\rho} & \times & \times \\ \times & \times & M_{C_b} & \times \\ \times & \times & \times & M_{C_s} \end{bmatrix} \begin{bmatrix} \dot{v}_t \\ \dot{v}_r \\ \dot{\sigma}_r \\ \dot{\sigma}_t \end{bmatrix} = \begin{bmatrix} \times & \times & \times & -D_1^\top \\ \times & \times & -D_2^\top & -D_0^\top \\ \times & D_2 & \times & \times \\ D_1 & D_0 & \times & \times \end{bmatrix} \begin{bmatrix} v_t \\ v_r \\ \sigma_r \\ \sigma_t \end{bmatrix} + \begin{bmatrix} \times & B_F \\ B_T & \times \\ \times & \times \\ \times & \times \end{bmatrix} \begin{bmatrix} \pi_{\partial,1} \\ \pi_{\partial,2} \end{bmatrix},
$$

$$
\begin{bmatrix} y_{\partial,1} \\ y_{\partial,2} \end{bmatrix} = \begin{bmatrix} \times & B_T & \times & \times \\ B_F & \times & \times & \times \end{bmatrix} \begin{bmatrix} v_t \\ v_r \\ \sigma_r \\ \sigma_t \end{bmatrix}.
$$

(D.15)

The mass matrices $M_{\rho h}$, $M_{I_\theta}$, $M_{\boldsymbol{C}_b}$, $M_{C_s}$ are computed as

$$
\begin{aligned}
M_{\rho A}^{ij} &= \left\langle \varphi_{v_t}^i,\, \rho A \varphi_{v_t}^j \right\rangle_\Omega, & M_{C_b}^{pq} &= \left\langle \varphi_{\sigma_r}^p,\, C_b \varphi_{\sigma_r}^q \right\rangle_\Omega, \\
M_{I_\rho}^{mn} &= \left\langle \varphi_{v_r}^m,\, I_\rho \varphi_{v_r}^n \right\rangle_\Omega, & M_{C_s}^{rs} &= \left\langle \varphi_{\sigma_t}^l,\, C_s \varphi_{\sigma_t}^s \right\rangle_\Omega,
\end{aligned}
$$

(D.16)

where $i, j \in \{1, N_{v_t}\}$, $m, n \in \{1, N_{v_r}\}$, $p, q \in \{1, N_{\sigma_r}\}$, $l, s \in \{1, N_{\sigma_t}\}$. Matrices $D_1$, $D_2$, $D_0$ assume the form

$$
\begin{aligned}
D_1^{lj} &= \left\langle \varphi_{\sigma_t}^l,\, \partial_z \varphi_{v_T}^j \right\rangle_\Omega, & D_0^{rn} &= -\left\langle \varphi_{\sigma_t}^r,\, \varphi_{v_r}^n \right\rangle_\Omega. \\
D_2^{pn} &= \left\langle \varphi_{\sigma_r}^p,\, \partial_z \varphi_{v_r}^n \right\rangle_\Omega,
\end{aligned}
$$

(D.17)

Vectors $B_F$, $B_T$ are computed as ($i \in 1, N_{v_t}$ and ($m \in 1, N_{v_r}$)

$$
B_F^i = \varphi_{v_t}^i(b), \qquad B_T^j = \varphi_{v_r}^j(b).
$$

(D.18)

### D.3   Control by Neural Approximators and MSL

Due to invertibility of the mass matrix we can reduce the above equation to a controlled linear system representing the discretized dynamics of the boudary-controlled Tymoshenko beam

$$
\begin{aligned}
\dot{z}(t) &= A z(t) + B \pi_\partial(t) \\
y_\partial(t) &= C z(t)
\end{aligned}
$$

(D.19)

with

$$
z = \begin{bmatrix} v_t \\ v_r \\ \sigma_r \\ \sigma_t \end{bmatrix}, \quad \pi_\partial = \begin{bmatrix} \pi_{\partial,1} \\ \pi_{\partial,2} \end{bmatrix}, \quad y_\partial = \begin{bmatrix} y_{\partial,1} \\ y_{\partial,2} \end{bmatrix}
$$

(D.20)

and

$$
A = \begin{bmatrix} \times & \times & \times & -M_{\rho A}^{-1} D_1^\top \\ \times & \times & -M_{I_\rho}^{-1} D_2^\top & -M_{I_\rho}^{-1} D_0^\top \\ \times & M_{C_b}^{-1} D_2 & \times & \times \\ M_{C_s}^{-1} D_1 & M_{C_s}^{-1} D_0 & \times & \times \end{bmatrix},
$$

$$
B = \begin{bmatrix} \times & M_{\rho A}^{-1} B_F \\ M_{I_\rho}^{-1} B_T & \times \\ \times & \times \\ \times & \times \end{bmatrix}, \quad C = \begin{bmatrix} \times & B_T & \times & \times \\ B_F & \times & \times & \times \end{bmatrix}
$$

(D.21)

We consider a parametrization $u_{\partial,\theta}$ with parameters $\theta$ of the boundary controller $\pi_\partial$ via a multi–layer perceptron. The neural network controller $\pi_{\partial,\theta}$ takes as input the discretized state of the PDE $\pi_\partial(t) = \pi_{\partial,\theta}(z(t))$, $t \mapsto z \mapsto u_{\partial,\theta}$. We apply the MSL to the controlled system

$$
\dot{z}(t) = A z(t) + B \pi_{\partial,\theta}(z(t))
$$

Further details on the experimental setup and numerical results are given in Appendix E.3.

# E Experimental Details

**Experimental setup**  Experiments have been performed on a workstation equipped with a 48 threads AMD RYZEN THREADRIPPER 3960X a NVIDIA GEFORCE RTX 3090 GPUs and two NVIDIA RTX A6000. The main software implementation has been done within the PyTorch framework. Some functionalities rely on torchdyn (Poli et al., 2020b) ODE solvers and torchcde (Kidger et al., 2020b) cubic splines interpolation utilities for the *interpolated* version of the adjoint gradients.

**Common experimental settings**  In all experiments to setup the multiple shooting problem, we choose an evenly spaced discretization of the time domain $[t_0, t_N]$, i.e.

$$\forall n = 1, \ldots, N \quad t_n = t_{n-1} + \frac{1}{N}(t_N - t_0)$$

## E.1 Variational Multiple Shooting Layers

**Dataset**  We apply *variational multiple shooting layers* (vMSL) to trajectory generations of various dynamical systems. In particular, we consider the Van Der Pol oscillator

$$\dot{p} = q$$
$$\dot{q} = \alpha(1 - p^2)q - p$$

as well as the Rayleigh Duffing system

$$\dot{p} = q$$
$$\dot{q} = \alpha p - 2p^3 + (1 - q^2)q$$

We generate a dataset of 10000 trajectories by solving the above systems until $T = 1$. Each trajectory consists of 20 regularly sampled observations subject to additive noise $\epsilon$ where $\epsilon \sim \mathcal{N}(0, \Sigma)$, with $\Sigma$ not diagonal i.e state–correlated noise.

**Models and training**  Both vMSLs as well as Latent Neural ODE baselines are trained for Latent Neural GDEs are trained for 300 epochs with ADAM (Kingma and Ba, 2014). We schedule the learning rate using one cycle policies (Smith and Topin, 2019) where the cycle peak for the learning rate is $10^{-2}$, set to be reached at epoch 100. The encoder architecture is shared across all models as is defined as two layers of *temporal convolutions* (TCNs), followed by a linear layer operating on flattened features. Between each TCN layer we introduce a maxpool operator to reduce sequence length. We solve Neural ODEs with dopri5 solver with tolerances $10^{-4}$.

We experiment with both fw sensitivity MSL as well as zeroth–order MSL as vMSL decoders. In all cases, we perform a single iteration of the chosen forward method. The parallelized ODE solves apply a single step of *Runge–Kutta 4*. We note that vMSL *number of function evaluation* (NFE) measurements also include the initialization calls to the vector field performed by the coarse solver to obtain shooting parameters $B_0$. Fig. 12 provides visualizations for decoder samples (extrapolation) of all models compared to ground–truth trajectories while Fig. 13 displays the learned vector fields of both vMSL and Latent ODE model.

To train all models we set the output–space prior $p(\hat{x}) := \mathcal{N}(x, \sigma)$ with $\sigma = 0.1$.

## E.2 Optimal Limit Cycle Control via Multiple Shooting Layers

In the optimal control tasks we considered a simple mechanical system of the form

$$\dot{q}(t) = p(t) \atop \dot{p}(t) = \pi_\theta(q(t), p(t)), \quad z = [q, p].$$

evolving in a time span $[t_0, t_N] = [0, 10]$ and we fixed $N = 99$. The task was the one of stabilizing the state of different *loci* $S_d = \{z \in \mathcal{Z} : s_d(z) = 0\}$ by minimizing $|s_d(z(t))|$, $s_d : [q(t), p(t)] \mapsto s_d(q(t), p(t))$. Specifically, we chose the following *loci* of points

1.  $s_d(q(t), p(t)) = q^2(t) + p^2(t) - 1$  [unit circle]
2.  $s_d(q(t), p(t)) = \sqrt{(q(t) - \alpha)^2 + p^2(t)}\sqrt{(q(t) + \alpha)^2 + p^2(t)} - k$

**Extrapolation Samples of Trained Decoders**
**Van Der Pol oscillator**

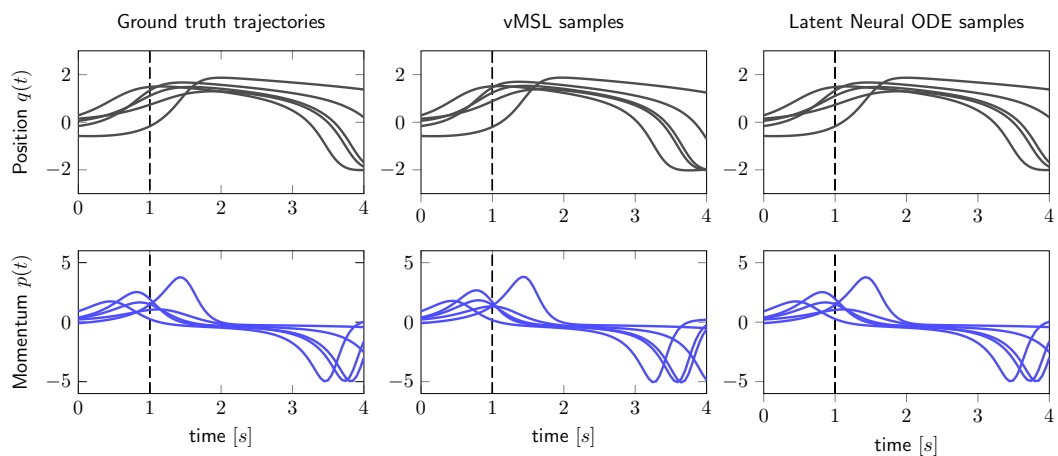

Figure 12: Samples of vMSLs and Latent Neural ODE baselines in the trajectory generation task on Van Der Pol oscillators. The samples are obtained by querying the decoders at desired initial conditions. The models extrapolate beyond $T = 1$ used in training.

**Variational MSLs & Latent ODEs**

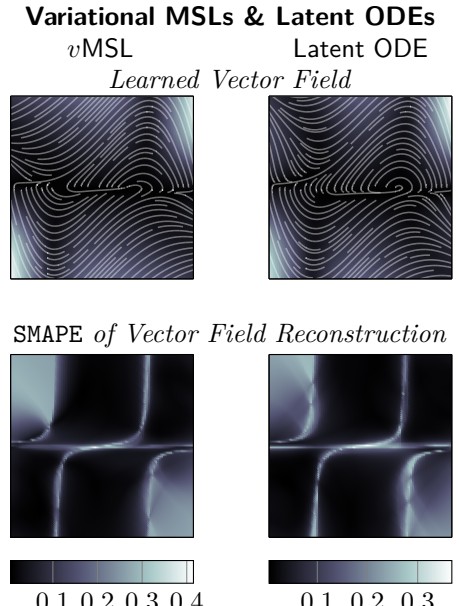

Figure 13: Learned vector fields by vMSL and Latent ODE decoders trained on noisy trajectories of the Van der Pol oscillator. vMSL models obtain the same result at a significantly cheaper NFE cost.

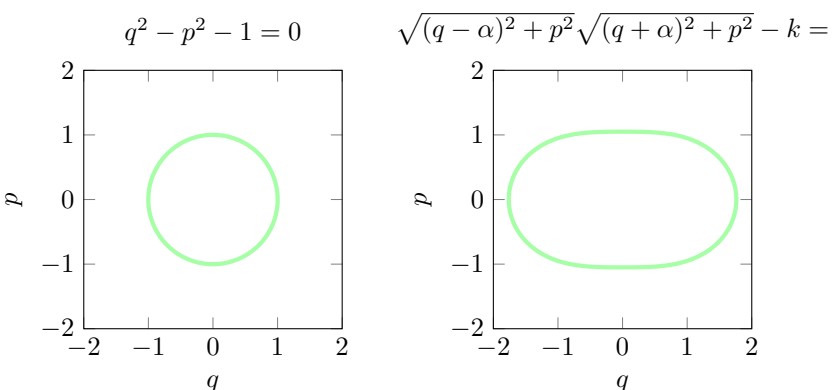

Figure 14: Desired *loci* in the state space, i.e. *limit cycles* to be created in the vector field through the control action $u_\theta(q(t), p(t))$.

across timestamps. The desired curves $s_d$ are displayed in Fig. 14.

We compared the performance of MSL with the one of a standard (sequential) Neural ODE trained with `dopri5` and `rk4` solver. The objective was to show that MSL can achieve the same control performance while drastically reducing the computational cost of the training.

**Models and training**    The loss function used to train the controlled was chosen as

$$\min_\theta \quad \frac{1}{2N|Z_0|} \sum_{j=0}^{|Z_0|} \sum_{n=0}^{N} \left| s_d \left( b_{n,j}^* \right) \right| + \alpha \left\| \pi_\theta(b_{n,j}^*) \right\|_1$$

$$\text{subject to} \quad B_j^* : g_\theta(B_j^*, z_0^j) = \mathbb{0}$$

$$z_0^j \in Z_0, \quad \alpha \geq 0$$

where $B_j^* = (b_{0,j}^* \ \cdots \ b_{N,j}^*)$. It penalizes the distance of trajectories from the desired curve as well as the control effort. In both the MSL and the Neural ODE baseline the controller $\pi_\theta(q, p)$ has been chosen as a neural network composed with two fully–connected layers of 32 neurons each and hyperbolic tangent activation. In the forward pass of MSL we performed a single iteration of the `fw sensitivity`–type algorithm. The parallelized ODE solver applies a single step of *Runge–Kutta 4* to each shooting parameter $b_n$. The backward pass has been instead performed with reverse–mode AD. At the beginning of the training phase, the shooting parameters $B_0^0$ have been initialized with with the sequential `dopri5` solver with tolerances set to $10^{-8}$, i.e. $B_0^0 = \{\tilde{\phi}_\theta(z_0, t_0, t_n)\}_n$. As described in the main text, $B^0$ has then been updated at each optimization step with the $B^*$ of the previous iteration to track the changes in the parameters $\theta$ and preserving the ability to track the "true" solution $\{\tilde{\phi}_\theta(z_0, t_0, t_n)\}_n$ with a single iteration of the Newton method (following the results of Theorem 1). The time horizon has been set to $[0, 10s]$ and we fixed $N = 100$ shooting parameters. The baseline Neural ODE has been instead trained with standard `dopri5` solver with tolerances set to $10^{-5}$ and the sequential `rk4` solver with $N$ steps over the time horizon.

It is worth to be noticed that both the *parallelized* `rk4` integration step of MSL and the *sequential* `rk4` integration in the Neural ODE baseline operates with the same step size of $0.1s$.

All models have been trained for 2500 epochs with a single batch of 2048 initial conditions $(q_0, p_0)$ uniformly distributed in $[-2, 2] \times [-2, 2]$ with ADAM (Kingma and Ba, 2014) optimizer and learning rate $10^{-4}$.

For the *circle* desired limit cycle, the training procedure has been repeated with different initial conditions and neural network initializations in a Monte Carlo Simulation of 50 runs. Further, at each training step of MSL we solved the forward system using `dopri5` with absolute and relative tolerances set to $10^{-5}$ to compute `SMAPE` with the current MSL solution across training iterations shown in Fig. 8. Similarly, throughout the training of the baseline Neural ODE we recorded the `NFEs` of the forward pass across iterations. We also repeated the training of each model recording the wall–clock time of every training iteration.

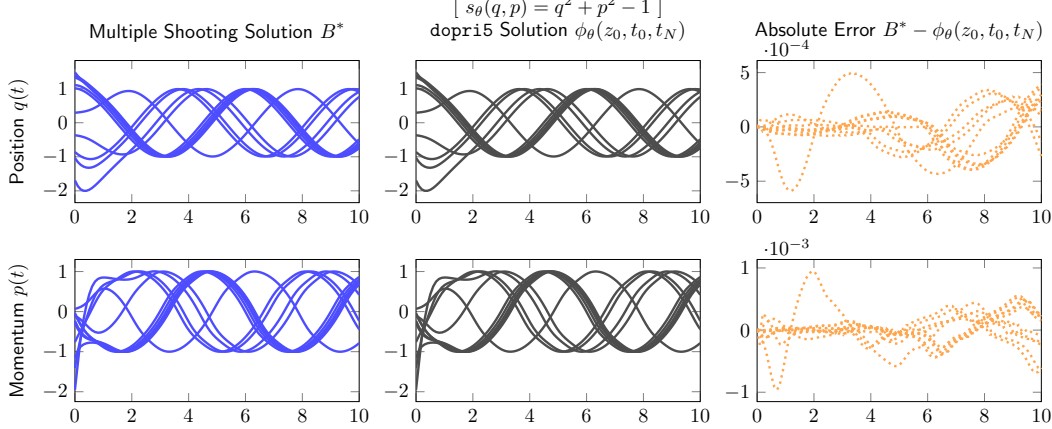

Figure 15: Trained MSL controller on the *circle* experiment. Comparison of the closed–loop trajectories obtained with MSL $B^*$ and the dopri5 counterpart.

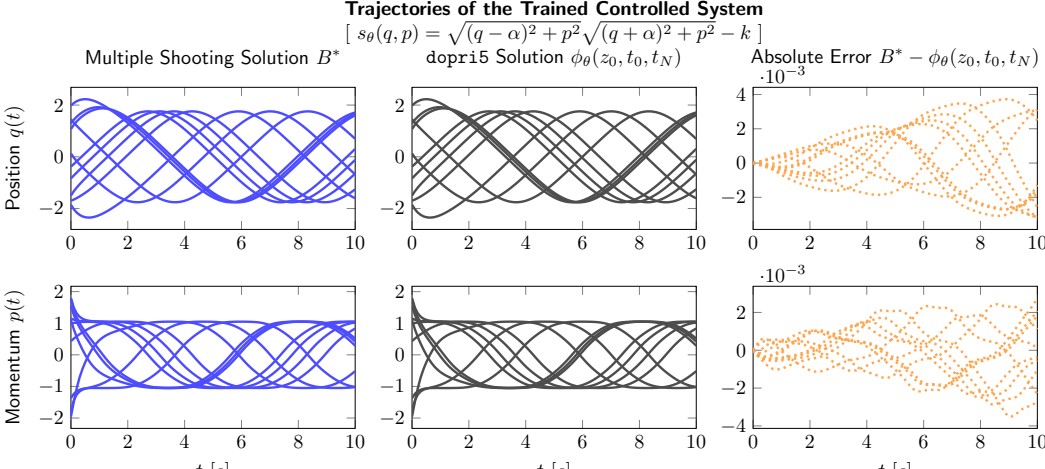

Figure 16: Trained MSL controller on the *circus* experiment. Comparison of the closed–loop trajectories obtained with MSL $B^*$ and the dopri5 counterpart.

**Analysis of results**  Figures 15 and 16 display the resulting trajectories of the trained MSL in the *circle* and *circus* control tasks. In particular, we compared the last MSL forward solution $B^*$ with the trajectories obtained with the accurate sequential solver with the trained $\pi_\theta$.

We also notice that the MSL and Neural ODE baseline converge to very similar controllers and closed–loop vector fields, as it is shown in Fig. 17.

In Fig. 18, we report the wall-clock times of each forward–backward passes across training iterations. It can be noticed how MSLs encompass sequential approaches with a 10x speedup compared to dopri5 (even though maintaining a similar accuracy in the solutions) and a 3x speedup w.r.t. the sequential rk4 solver with the same number of steps per sub–interval.

### E.3   Neural Optimal Boudary Control of the Timoshenko Beam

With this experiment we aimed at showing the scaling of fw sensitivity MSL to higher–dimensional regimes in a neural–network optimal control tasks. In particular, we wished to investigate if the acceleration property of *one–step* MSLs established by Theorem 1 holds when the system state has hundreds of dimensions.

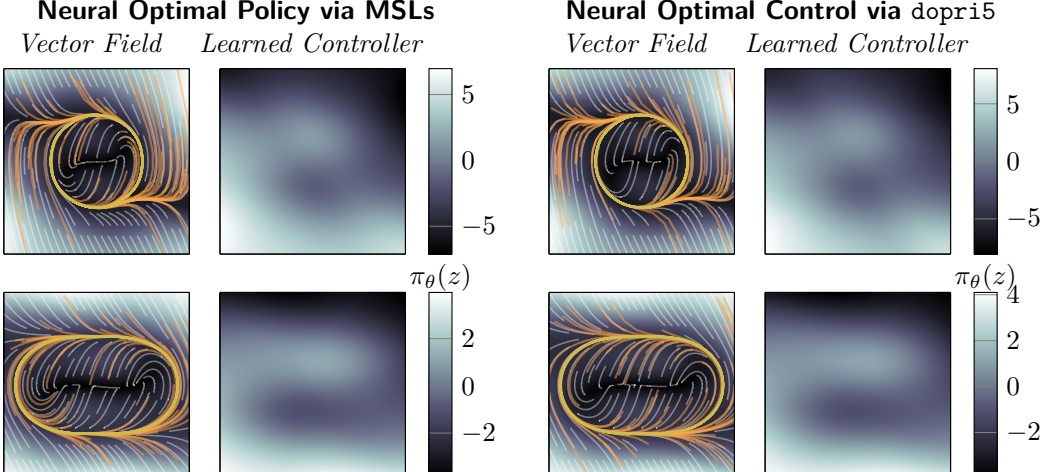

Figure 17: Comparison between the learned controllers and closed loop vector fields for the MSL and Neural ODE baseline, in different tasks.

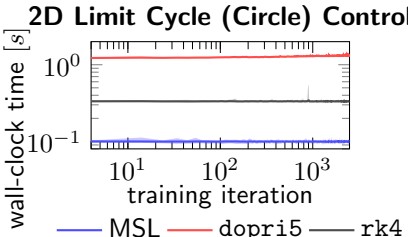

Figure 18: Wall-clock time of complete training iteration (forward/backward passes + GD update) for different solvers on the *circle experiment*

**Model and training**   We kept an identical training setup to the limit cycle control task of E.2. However, we chose a time horizon of $5s$ and we fixed $N = 500$ shooting parameters. We only compared the proposed MSL model to the sequential `rk4` as we empirically noticed how `dopri5` was extremely slow to perform a single integration of the discretized PDE (possibly due to the stiffness of the problem) and was also highly numerically unstable (high rate of *underflows*).

We implemented a software routine based on the `fenics` Alnæs et al. (2015) computational platform to obtain the finite–elements discretization (namely, matrices $A$ and $B$ in (D.19)) of the PDE given the physical parameters of the model, the number of elements, and the initial condition of the beam. We chose a 50 elements discretization of the Timoshenko PDE for a total of 200 dimensions of the discretized state $z(t)$ and we initialized the distributed state as $z(x, 0) = [\sin(\pi x), \sin(3\pi x), 0, 0]$.

Since the experiment focus was the numerical performance of MSL training compared to Neural ODE baselines, we considered a simple stabilization task where the cantilever beam had to be straight. For this reason we selected the following loss criterium

$$L_\theta = \frac{1}{N} \sum_{n=0}^{N} \left( \|b^*_{n,\sigma_r}\|_2 + \|b^*_{n,\sigma_t}\|_2 \right) + \alpha \|\pi_{\partial,\theta}(b^*_n)\|_1$$

being $b^*_{n,\sigma_r}$, $b^*_{n,\sigma_t}$ the portions of the shooting parameters corresponding to $\underline{\sigma}_r$ and $\underline{\sigma}_t$, respectively. The boundary controller was designed as a four-layers neural network with 16 neurons per layer, `softplus` activation on the first two hidden layers and hyperbolic tangent activation on the third.

**Analysis of results**   We report additional experimental results. Figure 19 displays the trajectories of the system with the learned boundary control policy. It can be seen how the displacements variables for each of the finite elements swiftly goes to zero (straight beam configuration) with zero

velocity proving the effectiveness of the proposed model. Finally, Fig. 20 shows the initial and final configurations of the finite elements over the spatial domain $x \in [0, 1]$.

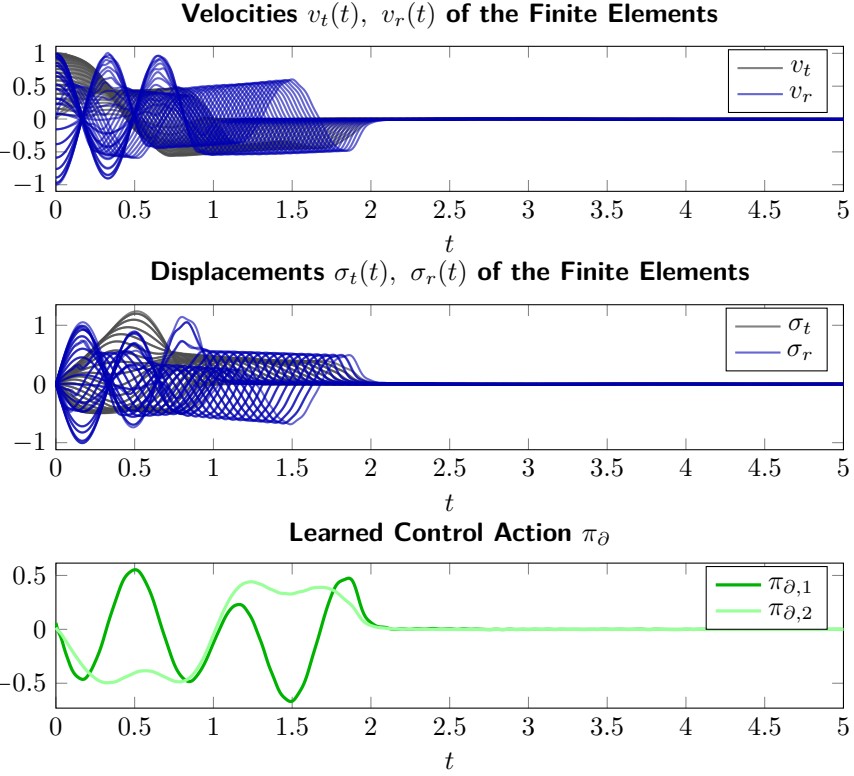

Figure 19: Trajectories of the finite elements states and learned control policy along the trajectory.

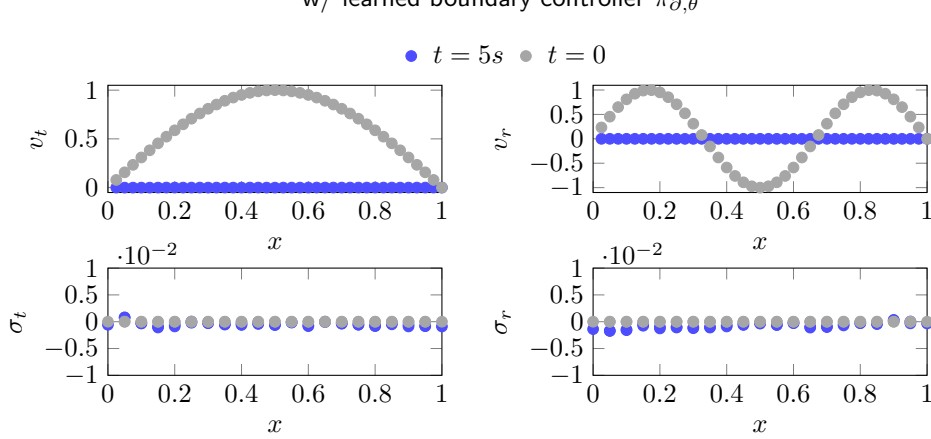

Figure 20: Initial and final (discretized) state of the controlled Timoshenko beam after training with MSL.

### E.4 Fast Neural CDEs for Time Series Classification

**Dataset** We consider sepsis prediction with data from the `PhysioNet` 2019 challenge. In particular, the chosen dataset features $40335$ variable length time series of patient features. The task involves

predicting whether patients develop sepsis over the course of their *intensive care unit* (ICU) stay, using the first 72 hours of observations. Since positive and negative classes are highly imbalanced, we report *area under the receiver operating characteristic* (AUROC) as task performance metric. For more details see (Kidger et al., 2020b), which contains the experimental setup followed in this work, and (Clifford et al., 2015) for more details on the dataset and task. The data split is performed according to (Kidger et al., 2020b) with 70% train, 15% validation and 15% test. The 70% split corresponds to 28233 time series, which in this experiment is taken as batch size to enable application of tracking MSLs relying on Theorem 1.

**Models and training** All model hyperparameters are collected from (Kidger et al., 2020b) for a fair comparison. We train a standard *neural controlled differential equation* (Neural CDE) and an equivalent Neural CDE solved with a zeroth–order MSL. Both baseline and MSL Neural CDEs use standard *reverse mode autodiff* to compute gradients. We train for 1000 epochs (here equivalent to iterations due to full–batch training) with a learning rate of $10^{-4}$ for AdamW (Loshchilov and Hutter, 2017) and with weight decay regularization of 0.03.