# OpenReview forum: "Differentiable Multiple Shooting Layers"
_NeurIPS.cc/2021/Conference — NeurIPS 2021 Poster_

### Official Review · Reviewer_cwaG · 2021-07-02

**Rating:** 6
**Confidence:** 5

**Summary:**

This paper presents a plug-in replacement for vanilla neural ODEs. The methodology relies on the multiple shooting idea, which divides a long sequence into subsequences and solves them concurrently.  The shooting variables are computed by Newton's root finding method (which requires one step per subsequence), which is then approximated by the "fixed point tracking" idea. The experiments show that in certain cases, the resulting technique leads to a reduction in the number of function evaluations and wall-clock time while achieving the same level of performance as neural ODEs.

**Limitations And Societal Impact:**

The authors addressed both the limitations and societal impact. However, I do believe the limitations should be experimentally illustrated.

**Main Review:**

This work brings an old but important idea into today's relevant literature. To the best of my knowledge, Newton's method for root finding has never been considered in the context of ODEs - so this is indeed a nice contribution. Utilizing forward sensitivities in the forward pass is also an interesting idea, which could impact future research as adjoints have been much more popular than sensitivities so far.

That being said, I believe the paper has certain clarity issues. Also, more (ablation style) experiments are needed to support/illustrate the claims. Finally, (although I might be mistaken in saying that), I believe the forward sensitivity derivation is wrong and should be revisited. I would be happy to increase my score if the authors address these points. Below are more detailed comments.

- The introduction section has clarity issues. I believe what the proposed framework does and why it is important are mixed. It would have been better to first clearly describe the problem, motivate why the problem is significant, and finally how the method addresses the described problem.
- Is proposition 1 correct? A modern derivation of sensitivities is given in [1]. If you look at eq. (8), you will see the term $\partial f/ \partial\theta$, which is missing in the submitted work (this term is missing because one needs to compute the total derivative whereas A.10 only computes the partial derivative by the chain rule.).
- Since the ODE solutions are computed numerically, this might impact the behavior of Newton's method. This should be demonstrated experimentally.
- Is the "one-step inference" method used in all experiments? If so, this should be clearly stated and a comparison between this approximation and the exact Newton solution should be given. Furthermore, in addition to the overall performance (demonstrated in 4.2), a simple experimental illustration showing the behavior of the bound is needed.
- Why are the NODE solver tolerances so high in sequence generation tasks? Comparison with default (if not tuned) values would be better. Also, I am not sure how to interpret Figure 6. Although NFE drops, the Jacobian computation might take significant time, making the method slower compared to latent ODE.
- The idea seems highly related to DEQs, which requires more explanation and possibly a conceptual comparison.
- More explanation/motivation for the "interpolated adjoints" is needed.

Minor comments:
- Figure 1 can be improved. In particular, the "RootFind" and b's can be shortly explained.
- Line 53: Is W a function of time? If so, this should be clearly explained. Otherwise, I'm having difficulty understanding the mapping $[0,T] \rightarrow R^{n_\theta}$.
- Formally defining $\phi_\theta(z,s,t)$ would help to quickly understand.
- l_x and l_y need clarification. If they refer to previous/next layers, a diagram would be better for illustrative purposes.
- Does the cross symbol denote
- Line 77: (3.1) before "theoretically".
- In line 55, $\phi$ has three arguments while it has only one in (3.2).
- Showing the derivation of (3.2) explicitly would be nice.
- Important references to sensitivities [2,3] are missing.
- Double "computed" in Figure 4 caption.
- Line 290: [CITE]

[1] Fröhlich, Fabian, et al. "Scalable parameter estimation for genome-scale biochemical reaction networks." PLoS computational biology 13.1 (2017): e1005331.
[2] Kokotovic, Petar, and James Heller. "Direct and adjoint sensitivity equations for parameter optimization." IEEE Transactions on Automatic Control 12.5 (1967): 609-610.
[3] Heinonen, Markus, et al. "Learning unknown ODE models with Gaussian processes." International Conference on Machine Learning. PMLR, 2018.

**Post-rebuttal comment:** Thanks for the very detailed response. The authors nicely addressed almost all the points I raised. I will raise my score to 6 "marginally above" since I believe the manuscript still suffers from certain issues:
- Reading the introduction, it is very difficult to clearly isolate the research problem, the gap in the literature, and the summary of the proposed approach.
- The following sections do not clearly describe the research problem either. What is the problem/posterior/likelihood/optimization objective? - - Obviously, the approach advertises itself as a replacement for neural ODEs. This is indeed a nice idea; however, for an outsider that quickly looks at the paper, immediately seeing the addressed problem is not really possible.
- Fig4 is definitely a much higher level explanation of the methodology and should precede all the figures except Fig1. So, I recommend moving the figure upwards and also refer it in the text.
- As a final suggestion, some of the technical material can be moved to the appendix and more verbal explanations of the concepts can be inserted.

**Time Spent Reviewing:**

6

---

> ### Author Response · Authors · 2021-08-11
> **Response pt.1 (cwaG)**
>
> We thank the Reviewer for the positive comments and detailed feedback. We include further experimental and ablation results as requested by the Reviewer in the anonymized link: https://gist.github.com/anonconfsub/04130d4ef5fd7751d6e4ca783f9c61b4. We agree with the Reviewer on the potential of forward sensitivity and Multiple Shooting in the context of Neural ODEs. We believe our approach to be particularly suitable to optimal control and model-based RL applications (see e.g. [12]).
>
> **Response to main comments**
>
> * We thank we Reviewer for the helpful suggestion. The introduction has been revised accordingly. MSLs are a Neural ODE alternative providing a flexible framework to parallelize certain forward-pass computations that are constrained to be sequential in Neural ODEs. We have focused on further highlighting how this purely sequential nature of Neural ODEs, addressed by MSLs, has limited their scaling and constrains them to be far below peak floating-point efficiency of current hardware accelerator, especially compared to other neural models.
> * After consulting reference [1] provided by the reviewer, we believe the forward sensitivity of Prop. 1 to be correct.
>
>     In our case, we are resorting to forward sensitivity to compute the total derivative of the *subflows* w.r.t. the *shooting parameters*, i.e. their initial conditions, and not the parameters $\theta$ of the vector field. Specifically, for the $n$-th subflow we have that $\phi_{\theta,n}(b_n) = z_n(t_{n+1}) = b_n + \int_{t_n}^{t_{n+1}} f_\theta(t, z_n(t))dt.$
>     Thus, by differentiating w.r.t. the shooting parameter $b_n$, we obtain
>
> $\frac{d \phi_{\theta,n}(b_n)}{db_n} = \frac{d z_n(t_{n+1})}{db_n} = \frac{d b_n}{d b_n} + \int_{t_n}^{t_{n+1}} \left[\frac{\partial f_\theta(t, z_n(t))}{\partial z}\frac{d z_n(t)}{d b_n} + \frac{\partial f_\theta(t, z_n(t))}{\partial t}\cancel{\frac{d t}{d b_n}} + \frac{\partial f_\theta(t, z_n(t))}{\partial \theta}\cancel{\frac{d \theta}{d b_n}}\right]dt$
>
> $=\mathbb I_{n_z} + \int_{t_n}^{t_{n+1}}\frac{\partial f_\theta(t, z_n(t))}{\partial z}\frac{d z_n(t)}{d b_n} dt$.
>
> The result follows by denoting $\mathsf D \phi_{\theta,n} = {d \phi_{\theta,n}(b_n)}/{db_n}$ and $\mathsf D f_\theta = \partial f_\theta /\partial z$ for compactness.
>     Therefore, the partial derivative $\partial f_\theta / \partial \theta$ is not expected to appear in the sensitivity ODE.
> * We agree about the importance of including a discussion on the effects of the numerical solver on the Newton update (3.2). Whenever a numerical ODE solver is used to obtain an approximation $\tilde\phi_{\theta, n}$ of the sub-flows $\phi_{\theta, n}$, the Newton method converges to the sequential solution $\tilde\phi_\theta$ of the same solver over the whole time horizon. For instance, if we applied $m$ steps of a 4th order Runge-Kutta scheme (`rk4`)in every sub--interval $[t_n, t_{n+1}]$, then the Newton iteration would eventually converge to the sequential `rk4` solution on $[t_0, t_N]$ with $mN$ steps. Note that this property is quite well-known in the numerical analysis literature (see e.g. [1]). There also exists a trade-off between computational cost over solutions accuracy for the choice of the number $N$ of shooting parameters $b_n$ (once fixed the number of ODE solver steps per sub-interval).
>     . While the sequential cost of MSL increase linearly with $N$, the overall solution accuracy in each time sub-interval $[t_n, t_{n+1}]$ should increase exponentially. With the number of ODE solver steps fixed, its step size $\Delta t$ is inversely proportional to $N$ and, for a $p$-th order solver, the global truncation error $E = o(\Delta t^p)$ is thus rescaled by $1 / N^p$.  We are open to suggestions about specific experiments that the Reviewer wish to see included in the revised version of the manuscript.
>
>
> * The "One-step inference" method is used in all experiments but the latent ODE model ($v$MSL). We retrained the one-step MSL on the optimal control task with the same settigs while keeping track of the error on the *one-step* solution error compared to the *full* MSL ($N$ steps of Newton iteration (3.2)) as well as `rk4` and `dopri5` sequential solutions. We report the results in Fig. 5 in the attached `gist` link. The mean absolute tracking error of the fixed point (`one-step MSL vs Full MSL`) is more than one order of magnitude lower than the one w.r.t to `dopri5`. Therefore the `one-step MSL` and `Full MSL` solutions are almost indistinguishable when compared to `dopri5`. In particular, they are almost as accurate as `rk4`. We believe that the discrepancy between the MSL methods and `rk4` are due to the integration errors of the forward sensitivities used to compute $\mathsf D\phi_{\theta, n}$ in (3.2). It can also be noticed how the tracking errors increase at the beginning of the training and slowly converge to a constant value as the norm of the loss gradient w.r.t the parameters decreases.  We believe such an analysis to be very informative and we will therefore extend it to all the other experiments as well. We thank the Reviewer for suggestion.
>
>     As requested by the Reviewer, we also performed a simple experiment to visualize the tracking-error bound. With different learning rates $\eta$, we performed one step of gradient descent to the same Neural ODE and recorded the tracking error. We repeated the experiment with several independent initializations of the Neural ODE and take the average (the code is reported in the gist link). We can see in Fig. 5 how it is possible to find a (minimum) constant $M>0$ such that $\|B^*_{p+1} - \bar B^*_p\|_2$ is always bounded by a quadratic function of the learning rate $M\eta^2$.
>
>
>   * Absolute and relative solver tolerances of $10^{-3}$ and $10^{-4}$ are usually sufficiently strict to ensure low solution error standard Neural ODE models for sequence and generative tasks [9], where dynamics are rarely stiff. Indeed, other works in this space also routinely use fixed-step solvers [10,11]. We note also that looser tolerances (i.e. bigger than $10^{-4}$, as chosen in our evaluation) is a regime that favours the latent Neural ODE baseline. Conversely, using stricter tolerances widens the gap of inference time between MSL and latent Neural ODEs, as shown in the additional experiment provided in the `gist`. In order to show MSL speedups, we have repeated the experiments on sequence generation and measured wall-clock time of MSLs and baselines. We have done so with different tolerances and using the same solver for MSL and baselines, as requested. Here, we measure the entire time taken by the model, which includes all operations, including root-finding updates.
>
>
> * We are grateful for the chance to discuss this point further due to its importance. The relation between DEQs and MSLs is the idea of casting the inference as a root-finding problem. DEQs seek the deep limit of weight-tied discrete neural nets $z_{k+1} = f_\theta(x, z_k)$. In particular, the inference aims at finding a stable fixed-point $z^*: z^* = f_\theta(x, z^*)$. In [2], the authors proposed to employ low-rank approximations of the Newton iteration (e.g. Broyden method) to seek directly a root of $g_\theta(x, z) = z - f_\theta(x, z)$. Although this procedure may output roots $z^*$ which are not *stable* (convergent) fixed-point of the discrete system (unless regularizers are used [3]), it consistently accelerates inference. In MSL we instead seek a root of $g(z_0, B) = B - \gamma_\theta(z_0, B)$ for guaranteeing the shooting parameters $B$ to be solutions of the underlying differential equation. In our case, thanks to the special structure of the Jacobian $\mathsf D\gamma_\theta(z_0, B)$, the Newton method admits an exact iterative update (3.2) in which we do not need to construct the full-Jacobian nor compute its inverse. Although (3.2) may appear "less parallel" than standard Newton (3.1) or Broyden methods, it actually reduces the sequential cost of the matrix-vector multiplication from quadratic $\mathcal O(n_z^2 N^2)$ to $\mathcal O(n_z^2(N-k))$. A paragraph has been added to Sec. 3 to offer such direct comparison between DEQs and MSLs.
>
>     Another interesting connection between MSL and DEQ is the result on fixed-point tracking and one-step inference which could be indeed applied to DEQs as well as **Reviewer 1tje** also pointed out. Finally, we acknowledge that a promising result aimed at accelerating DEQ was recently proposed in [8]. The basic idea is to "recycle" the approximated inverse Jacobian computed during the forward pass for back-propagation. We briefly mentioned the same idea in Appendix B.3 and we are confident that similar speedups can be also enjoyed by MSLs.
>
>
> * We agree with the importance of including a discussion on the choice of the interpolated adjoint to Sec. 3.2 which was currently missing due to space constraints. Backpropagation by interpolated adjoints is proven to be more accurate and robust to numerical errors, particularly for large or stiff ODEs where the standard adjoint may explode over long time horizons [4],[5],[6]. The main drawback of interpolated adjoints is the lower memory efficiency due to the *checkpoints* of the forward solution that have to be stored for the forward pass. However, in case of MSL interpolated adjoint result to be the natural choice as we can use as checkpoints the value of the shooting parameters $b_n$ at $t_n$ after the inference pass. We also notice that in the fixed-point tracking regime it may be faster to just back-propagate through the Newton solver rather than sequential interpolated adjoint.

---

> > ### Author Response · Authors · 2021-08-11
> > **Response pt. 2 (cwaG)**
> >
> > **Minor Points:**
> >
> > All typos, stylistic improvements and missing references has been addressed in the revised version of the paper version. In particular, we acknowledge the importance of [Petar and Heller] which is scarcely cited withing the Neural ODE field.
> >
> >
> > * **Q:** *Line 53: Is $\mathcal W$ a function of time? If so, this should be clearly explained. Otherwise, I'm having difficulty understanding the mapping $[0, T]\rightarrow\mathbb R^{n_\theta}$.*
> >
> >     As the reviewer points out, this is an important detail. The parameters $\theta$ of the vector field can be assumed constant in time or time-varying. The first-case is a clear special case of the second, which instead can represent time-varying functions. For the sake of generality, we therefore assume the parameter space $\mathcal W$ to be a space of smooth functions from $[0, T]$ to $\mathbb R^{n_\theta}$  (e.g. $L_2$ or $C^\infty$). However, optimizing a time-varying set of weights $\theta(t)$ corresponds to an optimization problem in infinite dimensions. Since optimizing in function space is computationally unfeasible, we then generally seek a discretization of the above. From a theoretical standpoint, the results proved in [7] for general Neural ODEs can be applied to MSL with time-varying parameters. We have added a formal discussion to time-variance in the parameters in supplementary material.
> >
> >
> > * **Q:** *Formally defining $\phi_\theta(z, s, t)$ would help to quickly understand.*
> >
> >     We thank the Reviewer for the suggestion. We agree that, in the current form, the def. of $\phi_\theta$ might be misleading. A more formal definition is strictly related to the *well-posedness* of the ODE (2.1) and has been erroneously taken for granted. In particular, if $f_\theta$ is smooth enough (Lipsichitz w.r.t $z$ and $\theta$ and uniformly continuous w.r.t $t$), then the IVP (2.1) admits a unique solution $z(t)$ defined on whole $[0, T]$. If this is the case there exists a mapping $\phi_\theta$ from $\mathcal{W}\times\mathcal Z\times [0,T]\times[0, T]$to the space of absolutely continuous functions $[0, T]\rightarrow \mathbb R^{n_z}$ such that for all $s,t\in[0, T]; s<t$ and $z_0\in\mathcal Z$, $z(t) = \phi_\theta(z_0, s, t)$ satisfies the IVP (2.1).
> >
> >
> > * **Q:** *$l_x$ and $l_y$ need clarification. If they refer to previous/next layers, a diagram would be better for illustrative purposes.*
> >
> >     The Reviewer is correct. $l_x$ and $l_y$ can be seen as input/output layers (e.g. the encoder or the readout in Variational MSLs). Nonetheless, we believe that the block diagram of Figure 4 is sufficient to portrait the overall model structure.
> >
> >
> > * **Q:** *In line 55, $\phi$ has three arguments while it has only one in (3.2).*
> >
> >     We thank the reviewer for spotting this incorrectness. As **Reviewer 1tje** pointed out, we forgot to define the sub-flow operator $\phi_{\theta, n}(b_n) := \phi_\theta(b_n, t_n, t_{n+1})$
> >
> >
> > * **Q:** *Showing the derivation of (3.2) explicitly would be nice.*
> >
> >     We agree with the Reviewer. In the current version of the manuscript we did not include it in the main text due to space constraints and forgot to include it in the supplementary materials. We have included all the steps from (3.1) to (3.2) to Appendix B.
> >
> > **Conclusions:**
> >
> > We gladly welcome further comments and questions by the Reviewer, and hope the current discussion addresses all technical concerns raised.
> >
> >
> > **References:**
> >
> >
> > [1] M. Gander, *Time Parallel Time Integration*
> >
> > [2] S. Bai et al., *Deep Equilibrium models*
> >
> > [3] S. Bai et al., *Stabilizing Equilibrium Models by Jacobian Regularization*
> >
> > [4] J. Zhuang et al., *Adaptive Checkpoint Adjoint Method for Gradient Estimation in Neural ODE*
> >
> > [5] T. Daulbaev et al., *Interpolation Technique to Speed Up Gradients Propagation in Neural ODEs*
> >
> > [6] https://diffeqflux.sciml.ai/stable/ControllingAdjoints/
> >
> > [7] S. Massaroli et al., *Dissecting Neural ODEs*
> >
> > [8] Z. Ramzi et al., *SHINE: SHaring the INverse Estimate from the forward pass for bi-level optimization and implicit models*
> >
> > [9] Y. Rubanova *Latent ODEs for Irregularly-Sampled Time Series*
> >
> > [10] C. Yildiz et al., *ODE2VAE: Deep generative second order ODEs with Bayesian neural networks*
> >
> > [11] P. Kidger et al., *Neural Controlled Differential Equations for Irregular Time Series*
> >
> > [12] C. Yildiz et al., *Continuous-Time Model-Based Reinforcement Learning*

---

> > ### Comment · Reviewer_cwaG · 2021-08-27
> > **Response to author response**
> >
> > Hi,
> >
> > Thanks for the very detailed response. The authors nicely addressed almost all the points I raised. I will raise my score to 6 "marginally above" since I believe the manuscript still suffers from certain issues:
> >
> > - Reading the introduction, it is very difficult to clearly isolate the research problem, the gap in the literature, and the summary of the proposed approach.
> > - The following sections do not clearly describe the research problem either. What is the problem/posterior/likelihood/optimization objective? Obviously, the approach advertises itself as a replacement for neural ODEs. This is indeed a nice idea; however, for an outsider that quickly looks at the paper, immediately seeing the addressed problem is not really possible.
> > - Fig4 is definitely a much higher level explanation of the methodology and should precede all the figures except Fig1. So, I recommend moving the figure upwards and also refer it in the text.
> > - As a final suggestion, some of the technical material can be moved to the appendix and more verbal explanations of the concepts can be inserted.
> >
> > Also, I would like to elaborate on one of the points I raised above (please consider this as a minor comment that does not affect my decision): Newton's method is highly sensitive to stochastic gradients. Numerical approximations to ODE solutions are surely deterministic. Nonetheless, slight changes in the vector field might cause drastic discrepancies in the resulting state trajectory, which in turn might affect the gradient stochasticity (over optimization iterations). This could be possibly explored in a simple experiment where one can check the number of iterations needed for convergence when the vector field is stiff/non-stiff.
> >
> > Thanks.

---

> > > ### Author Response · Authors · 2021-08-28
> > > **Response (cwaG)**
> > >
> > > We thank the reviewer for participating in the discussion. We are glad you found our response detailed and grateful for the increase in score.
> > >
> > > We note that the remaining points listed as remaining concerns had been already fixed, since they are all related to style and presentation. We have not elaborated in detail on how they had been fixed in our first response, as we decided to focus instead on discussing more pressing technical details. To summarize, we have made an effort to further highlight the research problem tackled by this work: developing an alternative to neural ODEs that is parallelizable in time.
> > >
> > >  The MSL model defines an inference procedure that produces valid ODE solutions, and is thus decoupled from specific likelihood / optimization details as is the case for neural ODEs. The objectives are related to the application under consideration i.e. continuous normalizing flows and latent models applications of neural ODEs / MSLs involving a maximum likelihood or ELBO loss, or dynamical system control applications involving a control loss. Section 4 develops details related to such applications and the corresponding optimization objectives, whereas the rest of the paper discusses the inference model and the numerics underpinning it. We have made this distinction clear in the introduction. We have further decided to follow the reviewer's suggestion to include more intuitive verbal explanations, deferring some technical details to the appendix.
> > >
> > > We appreciate the reviewer for providing valuable comments aimed at improving the readability and clarity of our work. We realize the discussion phase is about to end. However, we believe all stylistic and presentation improvements can be effortlessly incorporated in the final version. As such, we'd be happy to consider any further modifications that the reviewer wishes to see in the paper.

---

### Official Review · Reviewer_7kd1 · 2021-07-15

**Rating:** 7
**Confidence:** 2

**Summary:**

This work proposes a novel neural model (Multiple Shooting Layer) that is a significant improvement over Neural ODEs. MSLs seek solutions of initial value problems via parallelizable root-finding algorithms which lead to -
* significant speedups in wall-clock inference time;
* significantly less Number of Function Evaluations (NFE);

The authors further used their algorithm for -
* Neural Optimal Control - In this experiment they demonstrate that MSLs lead to orders of magnitude of NFE savings as compared to Neural ODEs.
* Time series classification - In this experiment they demonstrate that MSLs converge an order of magnitude faster than Neural ODEs.

**Ethical Concerns:**

Nil

**Limitations And Societal Impact:**

Nil

**Main Review:**

Originality:
This work uses Multiple Shooting Methods (a well known technique to solve PDEs in parallel) to propose a novel architecture that can serve as a direct replacement for Neural ODEs. MSLs come with several advantages -
* Require significantly smaller number of NFEs;
* Faster wall-clock time;
which are clear improvements over Neural ODEs.

Quality:
The authors have demonstrated the effectiveness of MSLs by conducting the following experiments -
* Neural Optimal Control - In this experiment they demonstrate that MSLs lead to orders of magnitude of NFE savings as compared to Neural ODEs.
* Time series classification - In this experiment they demonstrate that MSLs converge an order of magnitude faster than Neural ODEs.

Clarity:
The submission is well written.

Significance:
The effectiveness of MSLs over Neural ODEs is noteworthy as demonstrated in Neural Optimal Control and Time Series Classification experiments.

-------
# Post Rebuttal Comments
I've read the concerns raised by the reviewers and the authors have responded to them convincingly. For this reason I'd like to stick to my original score of 7.


**Time Spent Reviewing:**

2 hours

---

> ### Author Response · Authors · 2021-08-11
> **Response (7kd1)**
>
> We thank the reviewer for the positive comments and feedback. We would be glad to answer further questions in the hope of raising reviewer's confidence.

---

### Official Review · Reviewer_1tje · 2021-07-19

**Rating:** 6
**Confidence:** 4

**Summary:**

This paper proposes multiple shooting layers that solves a neural ODE dynamical system in parallel by reformulating IVPs as BVPs. The authors discussed the basic update rules and compared potential forward/backward modeling schemes, such as forward sensitivity, parareal method, and the potential recycling of fixed points (i.e., shooting parameters). The experiments show the benefits of the MSL formulation of ODE flows, which may significantly help the scalability of these (good but incredibly slow) implicit networks.

**Limitations And Societal Impact:**

Yes, the authors have adequately addressed the limitations.

**Main Review:**

Overall, I think the paper is clearly written and tackles an important issue (parallelization of ODE flows; bridging DEQs and NODEs). The idea of applying direct multiple shooting to model neural ODEs is also quite novel. To summarize:
- Good presentation and clear intuition explanation. The paper analyzed and compared different forward/backward dynamics for training an MSL and provided both theoretical and empirical analysis for the efficiency. I checked most of the proof, which seem correct.
- Applying the idea of direct multiple shooting to accelerate neural ODE solving seems novel to me, and bridges the equilibrium networks with the ODE-flow-based methods.
- Relatively thorough set of experiments showing the advantage of MSLs. However, considering that neural ODEs have also been applied in other more conventional domains (e.g., image classification), it would be interesting to see how well and efficiently the MSLs can perform in those settings.

-----------------------------

I also have a few questions/feedbacks:

1. Although the Newton-based update rule (3.2) is straightforward, I feel the second half of Sec. 3.1 is a bit redundant. For example, if I simply change (3.2) to $b_{n+1}^{k+1} = \phi_{\theta, n}(b_n^k)$, I would've arrived at the same conclusion in terms of "finite-step convergence". This will in practice make a difference because of 1) parallelism; and 2) the parareal method with coarse solvers; but at least theoretically I feel the time-iteration propagation discussion is a bit useless. Moreover, I don't think the notation $\phi_{\theta,n}$ is formally defined in the paper (and it took me a while to understand).

2. One discussion that is interesting but missing in the main text is the initialization $B^0$. How are these initializations picked, and how sensitive are the MSL convergences to them?

3. The authors suggest that unlike in deep equilibrium networks where one has to rely on either full vJP or low-rank quasi-Newton approximation to compute $(I-\mathsf{D}\gamma_\theta)^{-1}$, we can exploit the known structure of $D_\gamma$ and explicitly write out the inverse by a finite expansion $\left(I - \mathsf{D}\gamma_\theta(B^*)\right)^{-1} = I + \sum_{n=1}^N [\mathsf{D}\gamma_\theta(B^*)]^n$. However, besides the nilpotency, such usage based on Neumann series also require that the spectral radius of $\gamma_\theta$ to be $<1$. Is this generally a problem in practice?

4. The fixed-point tracking is indeed a great and interesting benefit of the fixed-point-based formulation (which applies also to works like DEQs, I presume?), but as the authors mentioned, they are also limited to the full-batch training mode. But in practice, unless we are dealing with an extremely small dataset, I don't feel this can be generally applied?

5. Is there any caveat that the authors find for using the zero-order (parareal) method coarse solver? I'm curious if this actually creates a problem when the underlying dynamical systems is quite involuted.

6. A major disadvantage of the approach is that, although the current experiments suggest acceleration, it is mainly because Neural ODEs are too slow and too small, not because MSLs are fast. For example, although Fig. 6 shows a significant reduction in NFEs, this is mainly a blessing of parallelization and we still have to undergo several root-solving iterations. If I'm understanding correctly, this means the actual FLOPs used for computation is actually larger, not smaller, than Neural ODEs. Therefore, will this acceleration benefit gradually diminish as we compare MSLs and NODEs on very large-scale and high-dim problems where parallelization yields only little gain?

**Minor points**:
- L137: $b_{n+1}^{b+1}$ should be $b_{n+1}^{k+1}$.
- L175: $\eta_k$ should be $\eta_p$.
- L290: There is a "[CITE]".

--------------------
--------------------

#### Post-rebuttal

I'd like to thank the authors for the response and agree that fixed-point tracking is an important direction to study for the implicit models. I can imagine how this benefit could enable implicit models to surpass/replace explicit models in many settings (e.g., where adjacent batches are correlated, such as in temporal sequences). I also appreciate the clarification on parallelism and FLOPs.

**Time Spent Reviewing:**

6

---

> ### Author Response · Authors · 2021-08-11
> **Response (1tje)**
>
> We thank the reviewer for the detailed feedback and positive comments. Before elaborating on the points raised, we would also like to share that during development of MSLs we had also performed experiments outside of optimal control and time series, including image classification. We include these in the anonymized link: https://gist.github.com/anonconfsub/04130d4ef5fd7751d6e4ca783f9c61b4. The results are consistent across domains, with MSLs matching Neural ODE performance with wall-clock time speedups.
>
> For our main submission we chose to focus on other domains (optimal control and time series) where Neural ODEs and CDEs have been shown to perform well w.r.t other classes of models.
>
> **Answers to main questions/feedback**
> 1. We agree about the redundancy of the finite-step convergence paragraph (straighforward) and the time-iteration propagation of Figure 2 and we moved these to Appendix B.1.
>     We further thank the reviewer for noticing the missing formal definition of $\phi_{\theta,n}$. With $\phi_{\theta,n}$ we refer to the *sub-flow* operator $\phi_{\theta,n}:\mathbb R^{n_z}\rightarrow\mathbb R^{n_z};b_n\mapsto \phi_{\theta,n}(b_n):=\phi_\theta(b_n, t_n, t_{n+1})$.
>
>
> 2. We have extended our discussion on the initialization of $B^0$ outside the fixed-point tracking regime. Similarly to DEQs, by relying on Newton method (or its zero-order apprimation), MSL is indeed sensitive to the choice of $B_0$ since only local convergence is guaranteed. Following classical literature on parallel-in-time and multiple-shooting methods [1],[2], we generally prescribe to initialize $B^0$ with a coarse Euler solution of the ODE. With an additional (sequential) cost of up to $\mathcal O(N)$ it is similar to the sequential cost of one extra Newton iteration. An alternative direction which we are currently exploring for future work is to reuse the previous activation as $B_0$ or compute it from $z_0$ with an auxiliary neural networks even outside the fixed-point tracking regime, since gradients w.r.t $B_0$ can also be obtained via implicit differentiation. These details have been added to the main text replacing the **finite-time convergence** paragraph.
>
>
> 3.  Generally speaking, the convergence of the Neumann series $\sum_{n=1}^\infty T^n, (T\in\mathbb R^{d\times d})$ requires boundedness of the spectral radius of $T$. Nonetheless, thanks to the nihlpotency of $\mathsf D \gamma_\theta$ all the terms of the series for $n>N$ are zero (as also pointed out in [3, Eq. 43]). Thus, the Neumann series reduces to a **finite** sum $\sum_{n=1}^N [\mathsf D\gamma_\theta]^n$ which can be iteratively computed without any particular assumption.
>
>
> 4. First, we confirm that the fixed-point tracking indeed applies also to most **implicit models**, including DEQs and optimization-like layers. We agree on the assumption of full-batch training
>     to be the main limitation of the result. While this can be readily applied to medium-scale datasets, it can certainly be more challenging, though not impossible, to apply to large datasets (on the scale common in vision or natural language). We note that many time series datasets of common use in deep learning are of sizes compatible with our fixed-point tracking approach. Indeed, we have shown that fixed-point tracking can be applied to sepsis classification to drastically reduce time-to-convergence, a task which features a dataset of common size in time series and other scientific machine learning domains. Generally speaking, it would also be interesting to experimentally evaluate for general DEQ models whether reusing the previous root as initial guess for the next batch of data leads to a reduction in NFEs compared to a random/zero initialization.
>
>     From a theoretical standpoint, we are currently exploring different research directions towards generality of the approach. Specifically, in the context of a follow-up on general *implicit models*, we are considering a fixed-point tracking bound for the stochastic regime (i.i.d sampling of inputs $x$ at each training iteration) by assuming a bounded sensitivity of $g_\theta$ w.r.t. to $x$ ($\exists M : \|\partial g_\theta/\partial x\|<M$) and requiring conditions on the training data distribution.
>
>
> 5. We thank the reviewer to for raising this interesting point. The `parareal` algorithm is generally very reliable and robust even in case of stiff ODE. Local convergence of the method is still guaranteed with a strong rate, as proved in [4]. [4] also shows several numerical results on stiff oscillators and the Lorenz chaotic ODE. Fig. 2 in the `gist` link displays the performance of our implementation of parareal on the Lorenz system compared to `Euler`, `rk4` and `dopri5` with low tolerances ($10^{-3}$). In our opinion, the only caveat concerns the choice of the coarse solver. Due to the added NFEs of the coarse solver $\psi$ needed to estimate $D\phi_{\theta, n}(b_n^k)(b_n^{k+1} - b_n^k)$, we found *Euler's method* to be the only coarse solver usable to obtain significant speedups in the general case.
>
>
> 6.  A primary objective of this work has been to provide a flexible blueprint where the degree of parallelizability can be controlled depending on the problem requirements. Neural ODEs are constrained to sequential computation in the time domain, whereas MSLs can tune how much computation is parallelized in time by choosing the number of shooting parameters. As with any parallel algorithm, scaling will occur up to a ceiling dependent on model, data size, and ultimately compute hardware and memory bandwidth.
>
>     In the application domains considered (including image classification), we have shown that the introduction of some degree of time parallelization in these models is beneficial in terms of wall-clock time, with speedups captured even on a single GPU. This suggests that Neural ODEs as a model class are far below peak floating-point utilization of generic accelerators.
>
>     We politely disagree with the statements that these models are small, noting also that the typical scale for continuous models is smaller than other implicit models. One of the causes of scaling limitations is precisely due to these models providing fewer parallelization opportunities (while being often *deeper* in their computational graph), which is precisely the weakness of Neural ODEs that MSLs attempt to alleviate. Several of the models trained, including MSLs for Neural CDEs and PDE control, are equipped with a parameter count equal to state-of-the-art models. For the image classification example, $20\cdot10^{6}$ parameters.
>
>     We agree, however, on the existence of some FLOP threshold (corresponding to parameters sizes significantly larger than what is common in continuous-depth models) beyond the peak floating--point performance in a roofline model of computation. Here, introducing further parallelism would not improve performance, since the algorithm is constrained by other factors such as memory bandwidth. However, as mentioned previously, most research and applications of continuous models are below such a threshold, and thus introducing further parallelism is often advantageous.
>
>     We note that relying only on FLOPs can be misleading in the analysis of parallel algorithms. We measure for wall-clock time as an aggregate of all computation done, including fine, coarse stepping, and all other operations in root-finding updates. We thank the reviewer for giving us the chance to elaborate on this important point. The discussion has been added to the manuscript.
>
> **Minor points**
> * We fixed all the typos and replaced the missing reference with [5]
>
> We would be glad to answer any further questions. We are always available to greatly expand upon specific aspects of the method.
>
> **References:**
>
> [1] J.L. Lions, Y. Maday, and G. Turinici, *A "parareal" in time discretization of PDE’s*
>
> [2] M. Gander, *Time Parallel Time Integration*
>
> [3] P. Chartier, B. Philippe, *A parallel shooting technique for solving dissipative ODEs*
>
> [4] M. Gander and E. Hairer, *Nonlinear Convergence Analysis for the Parareal Algorithm*
>
> [5] A. Macchelli and C. Melchiorri. *Modeling and control of the timoshenko beam. the distributed port hamiltonian approach.*

---

### Decision · Program_Chairs · 2021-09-27

**Decision:**

Accept (Poster)

**Comment:**

Bringing multiple shooting, time–parallel methods for ODEs to bear on improving inference and training of deep architectures based on Neural-ODE is a strong technical and conceptually novel contribution. As such, the paper may be expected to generate interest in both ML and Numerical methods (Differential Equations) communities. The reviews do ask for improved clarity in presentation and problem formulation in the final version.